# Substrate recognition mechanism of the endoplasmic reticulum-associated ubiquitin ligase Doa10

Kevin Wu[1,2,5], Samuel Itskanov[3,5], Diane L. Lynch[4], Yuanyuan Chen[1,2], Aasha Turner[1], James C. Gumbart [4] & Eunyong Park [1,2] ✉

Doa10 (MARCHF6 in metazoans) is a large polytopic membrane-embedded E3 ubiquitin ligase in the endoplasmic reticulum (ER) that plays an important role in quality control of cytosolic and ER proteins. Although Doa10 is highly conserved across eukaryotes, it is not understood how Doa10 recognizes its substrates. Here, we define the substrate recognition mechanism of Doa10 by structural and functional analyses on *Saccharomyces cerevisiae* Doa10 and its model substrates. Cryo-EM analysis shows that Doa10 has unusual architecture with a large lipid-filled central cavity, and its conserved middle domain forms an additional water-filled lateral tunnel open to the cytosol. Our biochemical data and molecular dynamics simulations suggest that the entrance of the substrate's degron peptide into the lateral tunnel is required for efficient polyubiquitination. The N- and C-terminal membrane domains of Doa10 seem to form fence-like features to restrict polyubiquitination to those proteins that can access the central cavity and lateral tunnel. Our study reveals how extended hydrophobic sequences at the termini of substrate proteins are recognized by Doa10 as a signal for quality control.

Selective degradation of misfolded and mistargeted proteins constitutes key pathways underpinning cellular protein homeostasis. In eukaryotic cells, the aberrant proteins are marked with polyubiquitin chains by E3 ubiquitin (Ub) ligases and subsequently degraded by the proteasome. The endoplasmic reticulum (ER) serves as a primary site for protein biosynthesis and maturation. Over one third of proteins translocate into the ER lumen or integrate into or associate with the ER membrane, including many proteins that are destined for other organelles[1–5]. To enable quality control of these proteins, the ER is equipped with a set of membrane-embedded E3 ligases that play central roles in the process known as ER-associated protein degradation (ERAD)[6–11]. The primary function of these E3 ligases is to recognize and polyubiquitinate aberrant proteins in the ER to enable their selective clearance. Moreover, through coordination with a AAA+ ATPase motor (Cdc48 in yeast and p97 in

mammals), ERAD-specific E3 ligases facilitate the removal of proteins from the ER lumen or membrane into the cytosol for proteasomal targeting, a process called retrotranslocation[12–16]. As the ER forms the most abundant membrane structure as well as a key biosynthetic hub, ERAD constitutes a vital component of protein quality control and regulated proteolysis in eukaryotic cells. Impairment of the ERAD machinery causes ER stress and dysfunction and is implicated in several human diseases[17,18]. However, the molecular mechanisms by which ERAD-specific E3 ligases mediate the protein quality control processes are incompletely understood.

Conceptually, ERAD can be classified into three categories depending on the topological location of substrate recognition with respect to the ER membrane: (1) the ER lumen, (2) the ER membrane, and (3) the cytosol. Respectively, these distinct ERAD pathways are

[1]Department of Molecular and Cell Biology, University of California, Berkeley, CA 94720, USA. [2]California Institute for Quantitative Biosciences, University of California, Berkeley, CA 94720, USA. [3]Biophysics Graduate Program, University of California, Berkeley, CA 94720, USA. [4]School of Physics and School of Chemistry and Biochemistry, Georgia Institute of Technology, Atlanta, GA 30332, USA. [5]These authors contributed equally: Kevin Wu, Samuel Itskanov. ✉e-mail: eunyong_park@berkeley.edu

often referred to as ERAD-L, ERAD-M, and ERAD-C[19-21]. ERAD-L substrates include misfolded lumenal proteins and membrane proteins with a misfolded lumenal domain, whereas ERAD-M substrates include those with misfolding in the intramembrane regions, and ERAD-C substrates are typically proteins with a misfolded cytosolic domain. In addition to misfolding, certain polypeptide segments and post-translational modification features of substrates also serve as degradation signals (degrons) for ERAD[22-24]. Generally, the different classes of substrates are recognized by distinct ERAD-specific E3 ligases. In fungal species, most ERAD-L/-M and ERAD-C substrates are handled by Hrd1 and Doa10, respectively, both of which belong to RING-type E3 ligases[25-27]. In addition to ERAD-C substrates, Doa10 also recognizes certain ERAD-M substrates[28]. Hrd1 and Doa10 are the two most conserved ERAD-specific E3 ligases across eukaryotic species including humans.

Doa10 (MARCHF6/TEB4 in metazoans and SUD1 in plants) is a large multi-spanning membrane protein residing in the ER and inner nuclear membranes[29] (Fig. 1a). Its first ~100 amino acids contain a RING-CH domain that enables its Ub ligase activity through an interaction with E2 Ub-conjugating enzymes. The RING-CH domain is followed by a transmembrane domain (TMD) containing 14 transmembrane segments (TMs), the structure and function of which are poorly characterized. Substrate polyubiquitination by fungal Doa10 requires two E2 proteins Ubc6 and Ubc7 (ref. 27). Ubc6 is a tail-anchored (TA) membrane protein in the ER membrane. Ubc7 is a soluble enzyme but localizes to the ER membrane through interaction with the single-spanning ER membrane protein Cue1 (ref. 30). While Ubc6 is involved in attachment of the first couple of Ub molecules to the substrate, Ubc7 is used for further elongation of the poly-Ub chain[31]. Doa10 is also shown to interact with Ubx2, a membrane protein

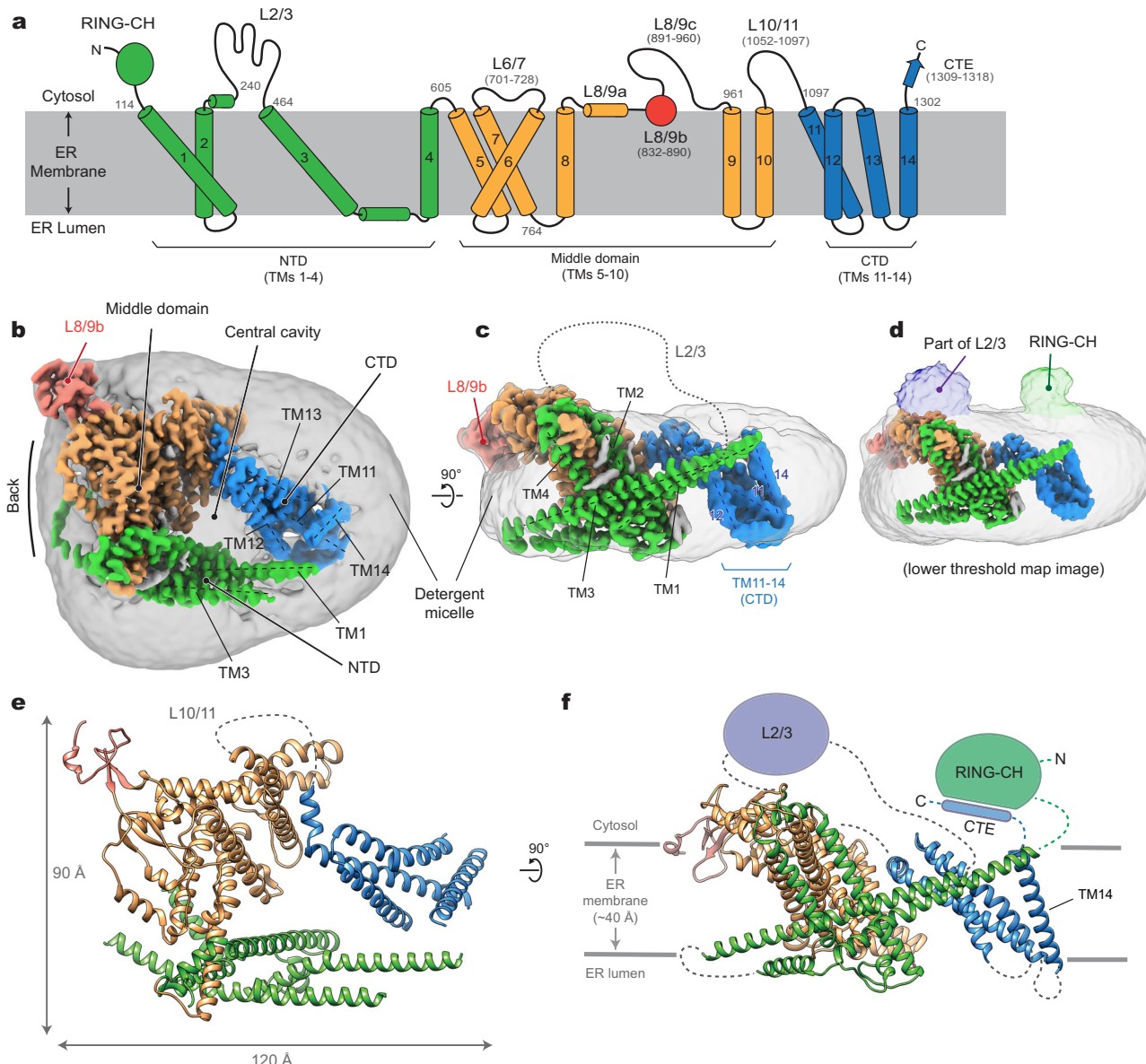

**Fig. 1 | Cryo-EM structure of *S. cerevisiae* Doa10 in detergent micelle.**
**a** Schematic diagram of the domain structure of *S. cerevisiae* Doa10. Helices are represented as cylinders (TMs 1-14 are numbered). Numbers in gray indicate amino acid residue number. **b** Cryo-EM density map of Doa10 viewed from the cytosol. Shown is a high-resolution protein density map overlaid with a lowpass-filtered detergent micelle density (gray). The color scheme is the same as in (**a**). **c** As in (**b**), but showing a side view along the membrane plane. **d** As in (**c**), but overlaid a density map with a lower surface threshold to show RING-CH and L2/3 features. **e**, **f** Atomic model of Doa10 based on the cryo-EM map. Views in (**e**) and (**f**) are equivalent to (**b**) and (**c**), respectively. Parts that are unresolved in cryo-EM were schematized.

that recruits Cdc48 to the ER, facilitating the extraction of substrate polypeptides from the ER membrane into the cytosol in the cases of membrane-associated substrates[32]. In addition to ER proteins, Doa10 can also polyubiquitinate soluble cytosolic proteins as its substrate recognition mainly occurs on the cytosolic side[33]. Furthermore, Doa10 has been shown to polyubiquitinate certain tail-anchored (TA) membrane proteins that are initially mistargeted to the outer mitochondrial membrane and then extracted by the Msp1 ATPase[34,35]. Thus, Doa10 recognizes a wide range of substrate proteins in cells.

Currently, the mechanism by which Doa10 recognizes its substrates and coordinates with its E2s for polyubiquitination is unclear. Although a short peptide called Deg1, which was derived from the yeast mating-type protein α2, has been identified as a Doa10-specific degron (Doa10 stands for Degradation of Alpha2 10) and often used as a model substrate in the studies of Doa10 (ref. [27,33,36,37]), it remains unknown how Doa10 recognizes Deg1. Interestingly, certain point mutations in the TMD of Doa10 have been found to alter the turnover rates of Deg1-fused protein substrates, suggesting a role of the TMD of Doa10 in polyubiquitination[38]. Moreover, it has been suggested that Doa10 itself can also dislocate certain transmembrane substrates from the membrane independently of substrate ubiquitination and Cdc48 (ref. [39]). Thus, the TMD of Doa10 may possess multiple functions including substrate recognition and retrotranslocation, but the mechanisms underlying these functions remain yet to be understood.

In this study, we present the results of our cryo-electron microscopy (cryo-EM), molecular dynamics (MD), AlphaFold2 modeling, and biochemical analyses of Doa10 from *Saccharomyces cerevisiae*. Our study reveals the highly unusual architecture of Doa10, where its TMD is arranged in a C-shape architecture with its central cavity filled with lipids. The conserved middle domain of Doa10 forms a lateral tunnel-like cavity, the entrance of which faces the central cavity and is open to the cytosol. Mutations in the tunnel impair degradation of Doa10 substrates, and MD simulations suggest that these mutations induce a collapse of the tunnel. Furthermore, we used photo-crosslinking to show that the lateral tunnel directly interacts with substrate polypeptides. Using membrane-anchored Deg1 substrates, we further show that a distance greater than 20–30 Å between Deg1 and the membrane anchor is required for efficient recognition of Deg1 by Doa10. This is likely because fence-like features of the N- and C-terminal portions of TMD restrict membrane proteins from accessing the central cavity and lateral tunnel. Our findings provide important insight into the mechanism by which Doa10 recognizes its substrates. Given the high degree of structural conservation, human MARCHF6 is also likely to use the same mechanism to recognize substrates.

## Results

### Cryo-EM analysis of S. cerevisiae Doa10

To gain structural insights into the mechanism of Doa10, we first determined a cryo-EM structure of *S. cerevisiae* Doa10. To purify Doa10, we inserted a cleavable green fluorescent protein (GFP) tag at the C-terminus of the chromosomal copy of Doa10 in yeast. We first tested purification of endogenous Doa10 by GFP affinity purification after solubilizing membranes with the lauryl maltose neopentyl glycol (LMNG) detergent. However, the yield and quality of samples were insufficient for structural analysis (Supplementary Fig. 1a, b). We therefore overexpressed Doa10 by replacing its endogenous promoter with a strong galactose-inducible *GAL1* promoter. The purified Doa10 protein in this approach yielded a largely monodisperse peak in size-exclusion chromatography and showed mainly as the full-length band on a SDS gel (Supplementary Fig. 1c, d). Cryo-EM images of the purified Doa10 sample displayed evenly dispersed particles, which produced well defined two-dimensional (2D) class averages (Supplementary Fig. 2a, b).

Single particle cryo-EM analysis of purified Doa10 yielded only one major class, which could be refined to a three-dimensional (3D) reconstruction at 3.2-Å overall resolution (Fig. 1b–d, and Supplementary Figs. 2c, 3 and Table 1). Most of the TMD were well defined, allowing us to build a reliable atomic model (Fig. 1e, f). However, we could not fully register distal parts of the last four TMs (i.e., TMs 11–14) due to their lower local resolution caused by the bending motions of the domain (Supplementary Figs. 3c, 4). We also note that the N-terminal RING-CH domain (residues 1–113) and the loop between TMs 2 and 3 (L2/3; residues 242–463) are only visible at low resolution due to high conformational heterogeneity (Fig. 1d). While we were conducting follow-up biochemical studies, AlphaFold2 was published[40]. An AlphaFold2 predicted model of yeast Doa10 agrees well with our experimental structure with a root mean square deviation (RMSD) of 2.7 Å over 835 aligned Cα atoms (Supplementary Fig. 5a, b). Since our cryo-EM structure cannot model the RING-CH domain and several loops, we also built a hybrid model combining our experimental model and high-confidence regions of the AlphaFold2 model (Supplementary Data 1).

### Overall architecture and structural features of Doa10

Doa10 exhibits highly unusual architecture. Overall, the TMD of Doa10 can be divided into the three subregions: the N-terminal domain (NTD), the middle domain, and the C-terminal domain (CTD), which are formed by TMs 1–4, 5–10, and 11–14, respectively (Fig. 1a–c). Viewed from the cytosol, they are arranged in C-shaped architecture (Fig. 1b). Consequently, Doa10 possesses a large enclosed 'central' cavity within its TMD. In our cryo-EM map, this cavity is filled with detergent and lipid molecules, suggesting that it would be occupied with lipids in the native membrane. Based on the low-resolution features, the RING-CH domain is placed directly above where TM1 and TM14 join at the tips of the C-shaped structure (Fig. 1d).

The TMD of Doa10 is also atypical in that many of its TMs are unusually long and highly tilted (Fig. 1). For example, TM3 is ~60 amino acids long and tilted by ~65° from the membrane normal. TMs 1, 5, and 9 are also ~50 amino acids long and tilted by more than 45°. A few other TMs (TMs 2, 6, and 11) are ~30–40 amino acids long, substantially longer than lengths of 20–30 amino acids for typical TMs. There are also multiple amphipathic domains. The segment between TMs 3 and 4 includes two amphipathic α-helices that would lie flat on the lumenal leaflet of the ER membrane. Part of the segment between TMs 8 and 9 (referred to as L8/9b and to be discussed later) forms a globular domain that would be partially embedded on the cytosolic leaflet of the ER membrane (also see Fig. 2d).

Another atypical feature of Doa10 is a relatively loose packing between its TMs. As a result, the TMD itself contains two sizable intra-TMD cavities each surrounded by TMs 1–5 (cavity 1) and by TMs 5–10 (cavity 2), respectively, and both are partly continuous from the central cavity (Fig. 2a, b; Supplementary Fig. 3e). Our cryo-EM structure shows that both cavities are occupied by lipids. Interestingly, cavity 2 is occupied with a triglyceride in addition to a phospholipid. The arrangement of TMs 5–10 suggests that these two lipids are unlikely to freely exchange with the bulk lipids, and therefore are likely trapped into the cavity during the folding of Doa10.

Among the poorly resolved regions in our cryo-EM are the N-terminal RING-CH domain and the C-terminal extension (CTE) following the TM14, which are expected to be in proximity to each other based on our cryo-EM structure (Fig. 1f). AlphaFold2 predicts that the CTE and part of the RING-CH domain co-fold into a two-stranded antiparallel β-sheet (Fig. 2c). Previous biochemical studies found that a truncation of CTE almost completely abolishes the degradation of Deg1 substrates[37]. This can be explained by a possible loss of proper functioning of the RING-CH domain without the CTE.

To test whether flexibility of the RING-CH domain observed in the cryo-EM structure could be an intrinsic property of Doa10, we ran 1-μs

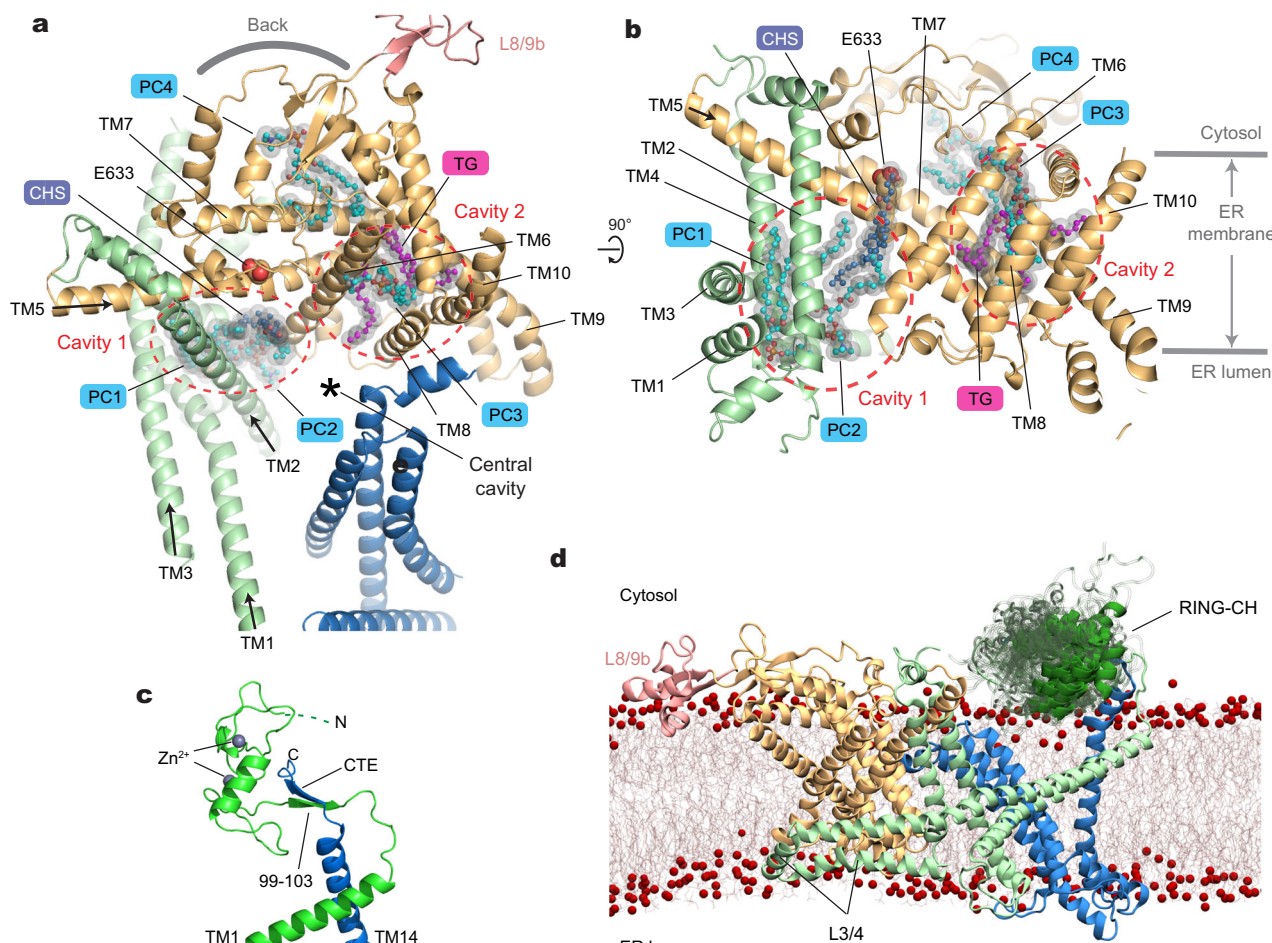

**Fig. 2 | Lipid molecules within the middle domain of Doa10 and the flexibility of the RING-CH domain. a** Lipid molecules entrapped in the interior of the TMD. The orientation of the Doa10 model (viewed from the cytosol) is rotated by -90° clockwise from Fig. 1b. **b** As in (**a**), but a lateral view from the central cavity to the middle domain. **c** An AlphaFold2 model of the RING-CH and CTE domains. Shown is a side view, equivalent to view in (**d**) and Fig. 1f. **d** Result of an MD simulation on WT Doa10 (side view). Positional flexibility of RING-CH is represented with multiple overlaid structures.

all-atom MD simulations on the hybrid model. Indeed, the results showed markedly larger mobility for the RING-CH domain compared to the TMD (Fig. 2d; Supplementary Fig. 4a–c and Movie 1). Throughout the course of the MD simulations, the CTE remained stably bound to the N-terminal RING-CH domain with the two β-strands forming 4 or 5 hydrogen bonds on average (Supplementary Fig. 4d, e). Thus, this observation suggests that a separation between TMs 1 and 14 is unlikely. Nevertheless, the cryo-EM structure indicates that CTD is relatively more mobile than the rest of the TMD (Supplementary Fig. 3c). Particle classification showed that the CTD wobbles by ~5° (Supplementary Fig. 4f–h), probably in part due to limited contacts between TMs 1 and 14. This flexibility would allow diffusion of lipid molecules between the bulk membrane and the central cavity of Doa10. On the other hand, integral (even single-spanning) membrane proteins would not easily enter the central cavity laterally through the seam between TMs 1 and 14 as the RING-CH:CTE contact would act as a barrier.

### Structural and sequence conservation of middle domain

Despite the highly interesting architecture of Doa10, the structure itself did not provide clear insights into the functions of the TMD. To better define functionally important features in Doa10, we mapped the amino acid conservation across Doa10 homologs onto the structure (Fig. 3a, b). This shows that the middle domain constitutes the most conserved region, whereas both NTD and CTD are substantially

variable. Specifically, within the middle domain, conserved TMs 5, 6, and 7, which together were previously referred to as the 'TEB4-Doa10 (TD)' domain[38], create a 'tunnel'-like cavity between a wedge formed by the TMs and the roof-like feature formed by the L6/7 loop, together with other parts of the middle domain (Fig. 3b). AlphaFold2 predicts that human homolog MARCHF6 also displays a highly similar structure in this region (Supplementary Fig. 5c–e). Among the most conserved residues in Doa10 are those amino acids lining the tunnel.

This lateral tunnel, which connects the central cavity and the back of Doa10, is largely level with the cytosolic leaflet of the ER membrane. However, the tunnel interior seems water-filled as the cavity is lined by a mixture of polar, charged, and hydrophobic amino acids. In fact, in our cryo-EM map, the tunnel entrance seems exposed to the cytosol with the detergent/lipid density in the central cavity tapering down around the entrance (Fig. 3c). Consistent with this, we also observed in the MD simulations that the tunnel interior is filled with many water molecules (Supplementary Fig. 6). On the other hand, the back of the tunnel seems at least partially blocked by a phospholipid in our cryo-EM structure (PC4 in Fig. 2a, b). Interestingly, one of the acyl chains of this lipid occupies part of the tunnel interior, contributing to hydrophobic surfaces in the tunnel.

Previous biochemical studies have shown that single point mutations of conserved E633 in the tunnel (Fig. 2b) affect the rates of substrate degradation to varying degrees depending on the amino acids it is mutated to (Asp or Gln) and the substrate[38]. This together

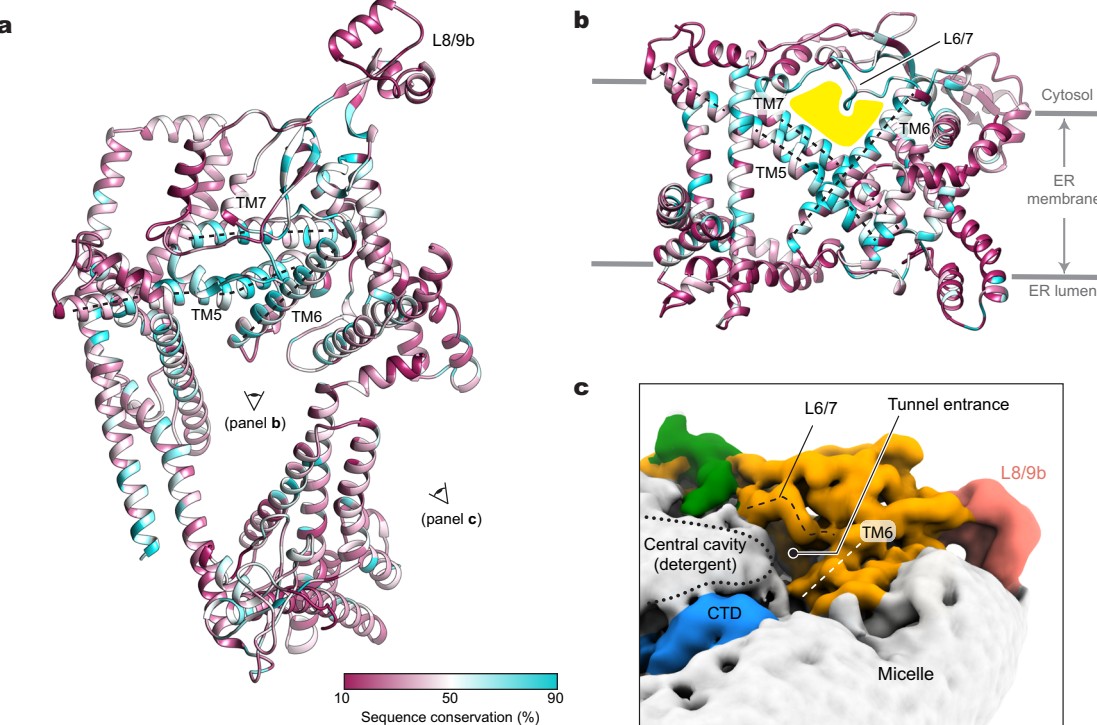

**Fig. 3 | Sequence conservation, lipid-filled central cavity, and water-filled lateral tunnel. a** Amino acid sequence conservation is mapped on the structure of Doa10. TMs 5 to 7 (referred to as the TD domain is indicated by dashed lines. The view (cytosolic view) is equivalent to that of Fig. 2a. **b** As in (**a**), but a lateral view from the central cavity to the middle domain. The lateral tunnel is highlighted in yellow. **c** A view into the entrance of the lateral tunnel. Shown in a lowpass-filtered cryo-EM map. The boundary of detergent features in the central cavity is indicated by a dotted line.

with the high sequence conservation and interesting structural features in this region prompted us to further investigate a possible role of the lateral tunnel in substrate recognition.

## Functional importance of the lateral tunnel in the middle domain

To test the effects of mutations in the tunnel on the function of Doa10, we first used the well-established uracil-dependent yeast growth assay[33]. In this assay, the Ura3 protein is expressed as a Deg1 degron fusion protein in an uracil-auxotrophic (*ura3Δ*) Doa10-null (*doa10Δ*) yeast strain together with an exogenous Doa10 variant. If the expressed Doa10 variant is functional, the growth of yeast in a medium lacking uracil (−Ura) becomes strongly inhibited due to efficient Deg1-Ura3 degradation (Fig. 4a). By contrast, a functionally defective Doa10 variant would allow a growth of the yeast as the Ura3 enzyme remains stable in the cytosol. To increase the dynamic range of the readout, we expressed Doa10 using two promoters with different strengths: *RET2* and *DOA10* promoters. While the *RET2* promoter expresses our exogenous Doa10 constructs at a level comparable to endogenous Doa10 in the WT strain, the expression level from the plasmid-borne *DOA10* promoter was substantially lower than this (Supplementary Fig. 7a), possibly due to the *DOA10* promoter used being partial or because exogenous Doa10 contained many synonymous codon replacements to facilitate molecular cloning of Doa10 (see "Methods"). We also noticed that a C-terminal GFP-tag somewhat reduces the expression level of Doa10 compared to a shorter peptide tag (ALFA-tag) (Supplementary Fig. 7a). Using different Doa10 expression levels in this way, we widened the dynamic range of phenotypic readouts of the assay.

When we tested for an E633Q mutant, we observed only a moderate growth rescue as observed previously[38] (Fig. 4b−d and Supplementary Fig. 7b), indicating a minor defect in Deg1-Ura3 degradation. The hydrophobic amino acid valine in the same position strengthened

this defective phenotype. Mutations on some other, but not all, amino acids lining the tunnel (such as S738V and Y742L) also exhibited similar defects (Fig. 4c, d). The double mutant E633V/S738V produced a somewhat stronger effect than E633V alone in the yeast growth assay (Fig. 4e), although the differences in the kinetics were less prominent than differences in the steady levels (Fig. 4f). Taken together, our data suggest that increasing the hydrophobicity in the tunnel interior can impair Deg1-Ura3 degradation.

Next, we tested whether altering the roof-like L6/7 feature affects Deg1-Ura3 degradation. A single point mutant on R710 or a larger replacement mutant (replacing Δ710−718 with a glycine/serine linker; Δ710−718::GS) caused no or relatively moderate impairment in Deg-Ura3 degradation (Fig. 4e and Supplementary Fig. 7c, d). A partially defective phenotype of the latter loop replacement mutant suggests that the native structure of L6/7 is not strictly required for substrate degradation. Strikingly, addition of a few hydrophobic amino acids (Δ710−718::GS+3Val and E713V/D714V) in L6/7 caused a stronger growth rescue. Consistent with this, cycloheximide chase experiments showed substantial stabilization of Deg1-Ura3 with Δ710−718::GS+3Val and E713V/D714V Doa10 (Fig. 4f and Supplementary Fig. 7e).

To understand the mechanism underlying the defects caused by the above mutations, we performed 1-μs MD simulations on mutant Doa10 containing E633V/S738V, Δ710−718::GS, Δ710−718::GS+3Val, or E713V/D714V (Fig. 4g−i; Supplementary Fig. 8 and Movie 2). In all mutants, we observed frequent collapses of L6/7 onto the wedge-shaped surface of the lateral tunnel formed by TMs 5−7 (Fig. 4j−l). Except for the Δ710−718::GS mutant, these collapses seems to be in part due to increased hydrophobic interactions between the L6/7 and the wedge-shaped surface of the tunnel. Consequently, these mutants exhibited a less solvent-accessible space in the tunnel compared to wild-type Doa10. We note that while the Δ710−718::GS mutation also collapsed L6/7, the interaction pattern was somewhat different from the other mutants tested: the replaced Gly/Ser loop of the

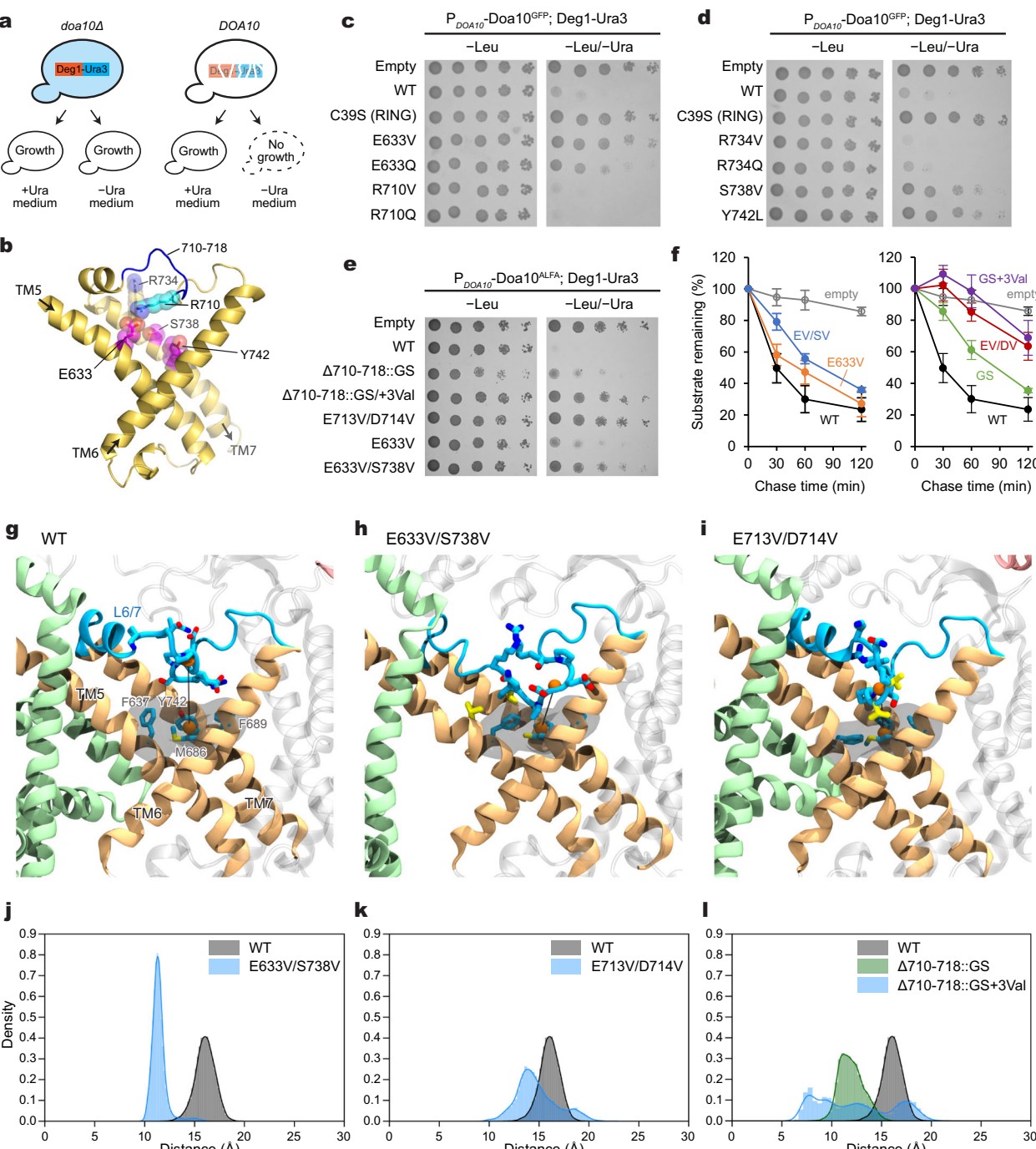

**Fig. 4 | Mutational analysis of the lateral tunnel of Doa10. a** Schematic diagram of Doa10-dependent yeast growth inhibition assay using Deg1-Ura3. **b** Structure of TMs 5 to 7 (TD domain). The view is similar to Fig. 3b. Positions of amino acid residues tested for the effects on Deg1-Ura3 degradation are shown with sticks and spheres (magenta: defects observed; cyan/blue: no substantial defects observed; see **c** and **d**). **c, d** Yeast growth inhibition assay with indicated Doa10 mutants (GFP-tagged Doa10 was expressed at a relatively low level from the *DOA10* promoter in a single-copy plasmid). **e** As in (**c**) and (**d**), but testing different mutants of Doa10 (ALFA-tagged). Note that a higher expression level of Doa10-ALFA compared to Doa10-GFP produces stronger growth inhibition. **f** Cycloheximide chase experiments on Deg1-Ura3 using indicated Doa10 mutants ('EV/SV' = E633V/S738V, 'GS' = Δ710-718::GS, 'GS+3Val'=Δ710-718::GS+3Val, and 'EV/DV' = E713V/D714V). Mean ± s.e.m. of three independent experiments. **g–i** Example snapshots of MD simulations for WT and indicated Doa10 mutants illustrating the collapse of the L6/7 loop. L6/7 are rendered in blue licorice, along with the sidechains for positions

710–718. The residues forming the hydrophobic wedge of the tunnel (F637, M686, F689, and Y742) are rendered in both gray semitransparent surface and licorice, with the center of mass (COM) of the loop residues 712–715 and the COM of the wedge residues rendered as orange spheres. Their separation is indicated with a black line. In these snapshots, the separation distances are 15.9 Å, 11.4 Å, and 14.0 Å for WT, E633V/S738V, and E713V/D714V, respectively. The mutated valine residues are rendered in yellow. See Supplementary Fig. 8 for GS and GS+3Val mutants. **j–l** The distance distribution was computed between the center of mass of the loop residues (712–715) and the hydrophobic wedge residues (F637, M686, F689, and Y742) for the wild type and mutant systems. The first 150 ns of each trajectory were removed to allow loop relaxation and the 2 replicas were combined. Data in (**c–e**) are representative of three independent experiments. Data in (**f**) are mean ± s.e.m. from three independent experiments. Source data are provided as a Source Data file.

Δ710–718::GS mutant mainly interacted with a polar surface formed by TMs 5 and 7 (Supplementary Fig. 8 and Movie 2), but in other mutants, the loop also interacted with a solvent-exposed hydrophobic surface formed around the bottom tip of the wedge (F637, M686, F689, and Y742) (Fig. 4g–l). If the lateral tunnel is to directly interact with the Deg1 peptide, such a prolonged interaction between the L6/7 and the tunnel's hydrophobic surface in the mutants would competitively inhibit binding of Deg1, which forms amphipathic helices. On the other hand, the somewhat weaker Deg1-Ura3 degradation defect of the Δ710–718::GS mutant might be explained by the observation that the hydrophobic surface is less stably occupied by its Gly/Ser loop.

### Probing interaction between Doa10 and Deg1 by photocrosslinking

Our structural and mutational analyses suggest the lateral tunnel in the middle domain potentially serves as a substrate-binding site. To probe a direct interaction between Doa10 and the Deg1 degron, we tested ultraviolet (UV) photocrosslinking between Doa10 and the Deg1 substrate in intact yeast cells. We first incorporated the non-natural photocrosslinkable amino acid p-benzoyl-L-phenylalanine (Bpa) into specific positions within the Deg1 sequence by amber stop codon suppression (Fig. 5a). Three out of four positions in Deg1 yielded crosslink adducts with Doa10 in a UV-dependent manner, suggesting a physical interaction between Deg1 and Doa10 (Fig. 5b).

Next, to examine an interaction between the middle-domain tunnel of Doa10 and Deg1, we incorporated Bpa into the tunnel interior and performed crosslinking with Deg1-Ura3 (Fig. 5c). Multiple positions in the tunnel interior showed crosslinking to the substrate (Fig. 5c). Positions 734 and 931 formed strong crosslinking, whereas some other positions (positions 738 and 906) showed weak crosslinking. Importantly, we did not observe obvious crosslinking when we introduced Bpa into the nearby cytosolically exposed surface (positions 231, 698, 916, 926, 944, and 948) (Fig. 5c and Supplementary Fig. 9), demonstrating that the observed crosslinking is not due to random collisions between Doa10 and the substrate. Importantly, the fact that multiple positions in the tunnel interior crosslink with Deg1-Ura3 (Fig. 5d, e) suggests that a part of the substrate polypeptide, most likely the Deg1 peptide, inserts into the lateral tunnel. Considering the confined dimensions of the tunnel, we tested this idea by fusing Trx1, a ~100-amino-acid-long globular protein, to the N-terminus of the Deg1-Ura3 substrate and performing the crosslinking experiment (Fig. 5f). Indeed, no crosslinking was detected between position 734 of Doa10 and the Trx1-fused substrate, suggesting an inability of Trx1-fused Deg1 to enter the tunnel. Additionally, we examined the effects of mutations in the tunnel and the L6/7 on the Bpa crosslinking. Consistent with the defects observed in growth assays (Fig. 4e, f), all tested mutations substantially decreased or abolished crosslinking between the tunnel and substrate (Fig. 5g). Collectively, these data show that Doa10's tunnel interior plays a vital role in facilitating Deg1 recognition through direct interaction.

### Role of the lateral tunnel in recognizing other substrates

In addition to Deg1, other known substrates of Doa10 include a fusion protein with a C-terminal peptide degron called CL1 (ref. 41) and certain TA membrane proteins, such as Sbh2 (ref. 28) and a truncated Pex15 (referred to as Pex15Δ30)[34,35]. All these substrates contain an amphipathic or hydrophobic segment at their C-terminus, which might interact with the lateral tunnel of Doa10 like Deg1. Thus, we tested whether the lateral tunnel is used to recognize these degrons as well. We first individually overexpressed three additional substrates, Ura3-CL1, mCherry-Sbh2, and mCherry-Pex15Δ30 together with Doa10 where Bpa was incorporated into its amino acid position 734 in the tunnel. UV treatment indeed produced crosslink adducts with these proteins as in the Deg1-Ura3 experiments (Fig. 6a, b). Again, the cytosolically-exposed amino acid position 926 of Doa10 showed no or

minimal crosslinking to these substrates, indicating a tunnel-specific nature of the crosslinks. We note that these substrates displayed lower crosslinking efficiencies than Deg1. This might be in part due to relatively lower expression levels of these proteins. In the cases of Sbh2 and Pex15Δ30, this could be additionally due to their limited cytosolic pools available to Doa10, considering that they would first need to be extracted from membranes into the cytosol before insertion into the lateral tunnel of Doa10 (see "Discussion").

To further test the involvement of the lateral tunnel, we examine the effects of tunnel-collapsing mutants (E633V/S738V and E713V/D714V) on the degradation of Sbh2 and Pex15Δ30 by measuring the fluorescence intensities of cells expressing mScarlet-fused constructs. The results showed that the steady-state levels of both mScarlet-Sbh2 and mScarlet-Pex15Δ30 were significantly elevated with these tunnel mutants compared to levels observed with WT Doa10 (Fig. 6c, d). Consistent with this, cycloheximide chase experiments also showed decreased degradation rates of these substrates with the tunnel mutants (Fig. 6e, f). However, we note that effects of two mutants were different between Sbh2 and Pex15Δ30. Sbh2 degradation is moderately affected by the strong E713V/D714V mutant in contrast to the cases of Pex15Δ30 (Fig. 6d) and Deg1 (Fig. 4e, f), whereas E633V/S738V showed rather minor effects on Pex15Δ30 (Fig. 6d, f). Collectively, these observations suggest that while the lateral tunnel is used for recognition of a variety of substrates, the specific interactions with the tunnel can vary depending on the substrate. In addition, certain substrates like Sbh2 might also utilize an alternative recognition mechanism, considering only partial defects observed with the tunnel mutants.

### Positions of E2

During the substrate polyubiquitination, the RING-CH domain of Doa10 is expected to interact with E2 proteins Ubc6 and Ubc7 to position the C-terminus of the Ub close to the substrate for ligation to occur. Because neither Ubc6 nor the Ubc7–Cue1 complex was included in our cryo-EM analysis, we generated AlphaFold2 models for Doa10 in complex with Ub and Ubc6 or Ubc7–Cue1 (Fig. 7a–f, Supplementary Fig. 10a, b). These models indeed predicted expected interactions among the RING-CH domain, E2 domain, and Ub. In addition, the models predicted that both Ubc6 and Ubc7–Cue1 associate with Doa10 through interaction with L8/9b of Doa10. Interestingly, Ubc6 and Cue1 are predicted to bind to opposite sides of L8/9b (Fig. 7f), making it possible for both Ubc6 and Ubc7–Cue1 to be simultaneously tethered to Doa10, while the E2 domains of Ubc6 and Ubc7 would engage with the RING-CH domain one at a time.

To biochemically probe the AlphaFold2-predicted interactions between L8/9b and Ubc6 or Cue1, we first truncated L8/9b from Doa10 (ΔL8/9b) and performed co-immunoprecipitation (co-IP) experiments. Although L8/9b is not a universally conserved feature among the Doa10/MARCHF6 proteins (for human MARCHF6, see Supplementary Fig. 5), the yeast growth assay indicated that a deletion of L8/9b largely abolishes Deg1 substrate degradation (Supplementary Fig. 10c, d), as expected for a loss of proper E2 interactions. Consistent with this, the co-IP experiment showed that ΔL8/9b substantially decreased co-purification of Cue1 (Fig. 7g). On the other hand, ΔL8/9b did not affect the amount of co-purified Ubc6, suggesting a possibility that copurified Ubc6 remained bound to Doa10 through an additional interaction site(s).

To further probe Cue1 and Ubc6 interaction with Doa10, we performed in-vivo UV-photocrosslinking experiments by incorporating Bpa into positions in Doa10 based on the AlphaFold2 models (Fig. 7h, i, Supplementary Fig. 10e). Strong crosslink adducts could be observed with multiple positions expected to be close proximity to Ubc6 or Cue1 (positions 928 and 949 for Ubc6, and positions 857 and 949 for Cue1), further providing the evidence for the Doa10-E2 interactions predicted by the AlphaFold2 modeling. However, the observations that position

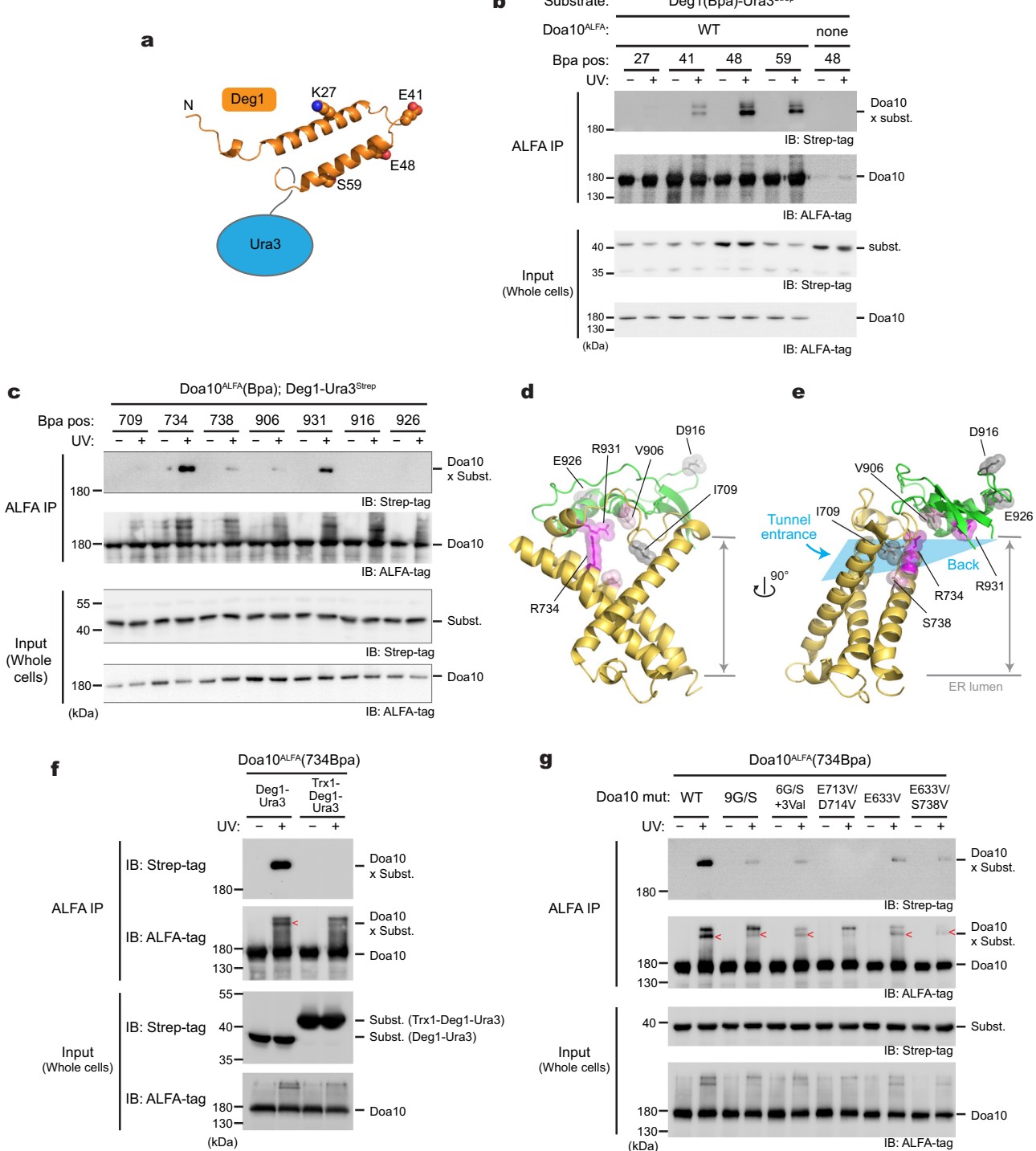

**Fig. 5 | Deg1 inserts into the lateral tunnel of Doa10. a** Schematic diagram of Deg1-Ura and the predicted secondary structure of Deg1 using AlphaFold2. **b** In-vivo photocrosslinking experiment using Bpa-incorporated Deg1-Ura and wild-type Doa10. IP, immunoprecipitation. **c** As in (**b**), but with Bpa incorporated into a specific position in Doa10 instead of Deg1. **d, e** Structure of the lateral tunnel (**d**, front view; **e**, side view), highlighting positions tested for crosslinking in panel

(**c**). Ribbons in yellow and green represent TMs 5–7 and L8/9c, respectively. Residues in magenta crosslink to Deg1-Ura3. **f** As in (**c**), but testing the effects of Trx1 fusion on crosslinking with 734Bpa of Doa10. **g** As in (**c**), but testing the effects of tunnel mutations on crosslinking with 734Bpa of Doa10. Data in (**b, c, f, g**) are representative of two independent experiments. Source data are provided as a Source Data file.

857 of Doa10 crosslinked to Ubc6 and that no crosslinking was observed between position 863 and Cue1 are somewhat inconsistent with the models.

While further structural investigations would be necessary to fully validate the AlphaFold2 models, an important implication of them is that, in the E2 and Ub-containing complexes, the C-terminal double-Gly tail of Ub would be positioned immediately (~10–15 Å) above the central cavity of Doa10 and in a close (~25 Å) distance from the tunnel entrance (Fig. 7a, b). A similar feature of Ub positioning with respect to human MARCHF6 could also be found from Alpha-Fold2 modeling with UBE2J2, a cognate E2 of MARCHF6 (Supplementary Fig. 10f, g).

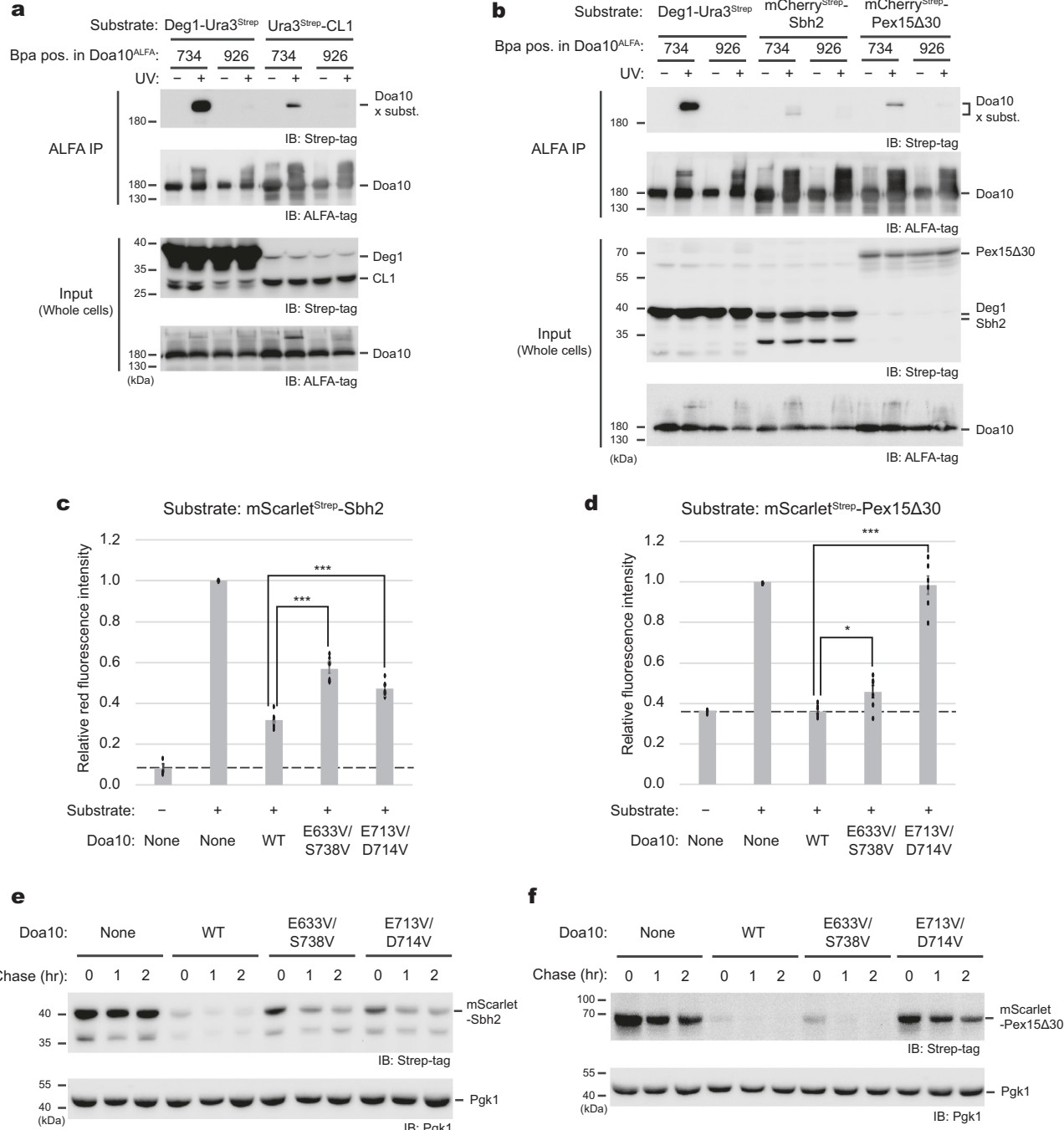

**Fig. 6 | Lateral tunnel as a general substrate-recognition site. a** In-vivo UV photocrosslinking experiment testing a direct interaction between the CL1 degron and lateral tunnel of Doa10. *p*-Benzoyl-L-phenylalanine (Bpa) was incorporated into either amino acid position 734 (tunnel interior) or 926 (cytosolic surface) of Doa10. Deg1-Ura3-2xStrep was used as a positive control. IP, immunoprecipitation. **b** As in (**a**), but testing mScarlet-Sbh2 and mScarlet-Pex15Δ30 as substrates. Note that these substrates also contain 2xStrep-tag immediately after mScarlet for immunodetection. **c** mScarlet fluorescence intensity measurements on yeast cells expressing mScarlet-Sbh2 (under the *TEF1* promoter) and Doa10 variants (under the *RET2* promoter). Where indicated, the substrate or Doa10 or both were omitted in constructing strains. Intensities were normalized with respect to the WT Doa10-expressing strain. Mean ± s.e.m. of 6 independent experiments. **d** As in (**c**), but testing mScarlet-Pex15Δ30 as a substrate. Mean ± s.e.m. of 7 independent experiments. **e, f** Cycloheximide chase experiments using mScarlet-Sbh2 (**e**) or mScarlet-Pex15Δ30 (**f**) and indicated Doa10 variants (strains identical to those in **c** and **d**). Pgk1 was used as a loading control. Data in (**a, b, e, f**) are representative of three independent experiments. Statistical significance was accessed by unpaired *t*-test (two-tailed). *\*p = 0.0091; \*\*\*p < 0.0001.* Source data are provided as a Source Data file.

To assess the dynamics of the E2 and Ub-bound RING-CH domains, we ran 1-μs all-atom MD simulations on both Ubc6- and Cue1-Ubc7-containing models (Supplementary Fig. 11). When we measured the distance between the double-Gly tail and the tunnel entrance, it fluctuated around 20 Å in the Doa10–Ubc6–Ub complex and around 25 Å in the Doa10–Cue1–Ubc7–Ub complex (Supplementary Fig. 11a, b). While the positional flexibility of the RING-CH domain remains much higher compared to the TMD in these simulations (Supplementary Fig. 11c, d), some reduction has been observed, particularly in the Doa10–Ubc6–Ub complex, compared to the

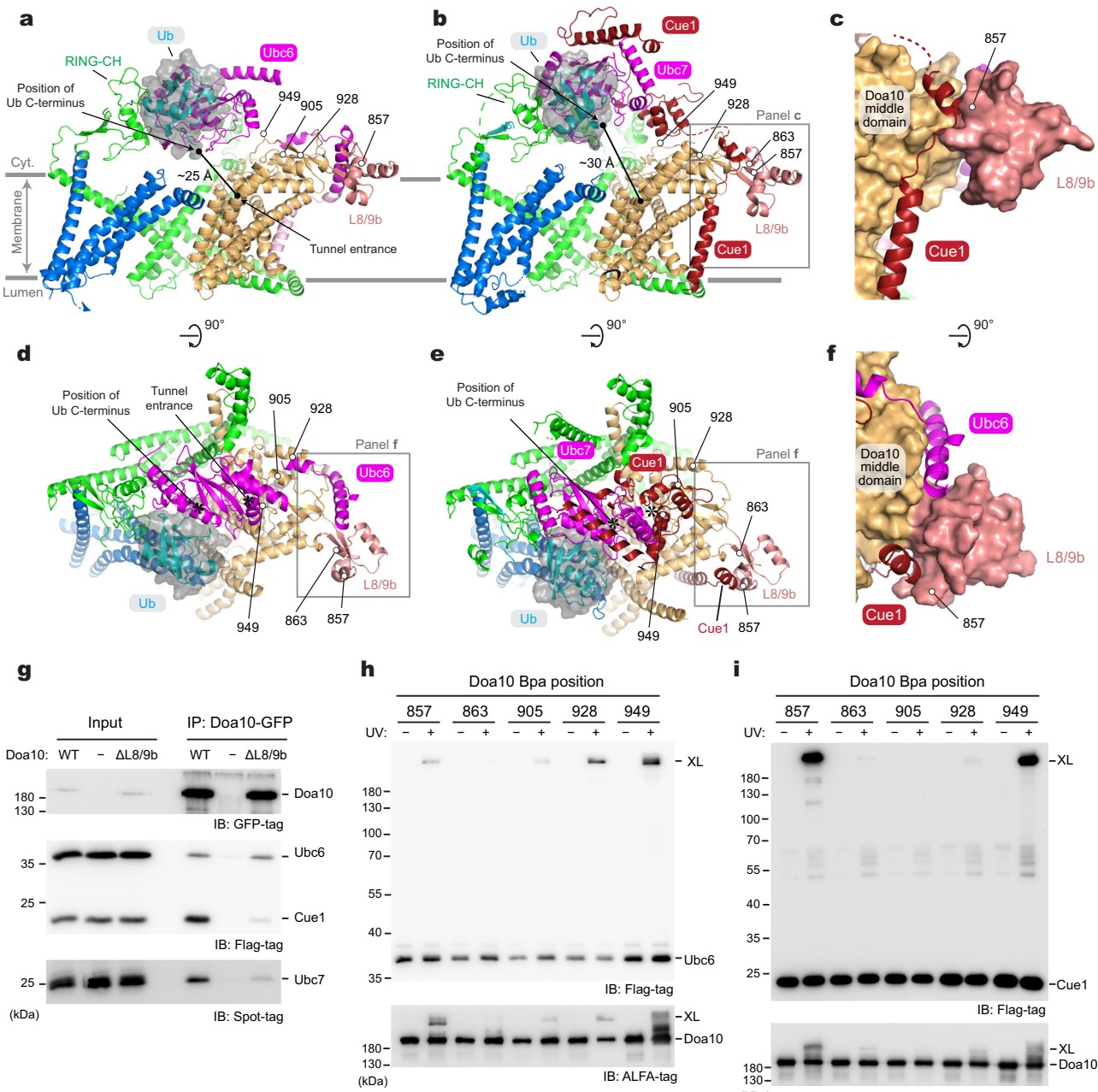

**Fig. 7 | Interaction between Doa10 and E2s and putative polyubiquitination site. a** AlphaFold2 model of Doa10 in complex with Ubc6 and Ub (a side view along the membrane plane). For Doa10, the RING-CH and N-terminal domains are shown in green, the middle domain in light brown, L8/9b in salmon, and the C-terminal domain in blue. Ubc6 is shown in magenta, and Ub is represented as cyan ribbon and semitransparent gray surface. Low-confidence (pLDDT <50) regions were now shown. We generated a total of five models, and all models were highly similar to one another with Cα RMSD ranging from 0.9 to 1.5 Å. **b** As in (**a**), but showing a model for the Doa10–Cue1–Ubc7–Ub complex. The area outlined with a gray box is shown in panel (**c**). We generated a total of five models, and all models were highly similar to one another with Cα RMSD ranging from 1.5 to 2.2 Å. **c** A view highlighting the interaction between Cue1 and L8/9b of Doa10 (Doa10 is in a surface representation). **d** As in (**a**), but showing the top (cytosolic) view. The area in a gray box is shown in panel (**f**). **e** As in (**b**), but showing the top (cytosolic) view. The area in a gray box is shown in panel (**f**). **f** A view highlighting the interaction of Cue1 or Ubc6 with L8/9b of Doa10 (composite of the two models shown in **d** and **e**). **g** Co-immunoprecipitation of Ubc6, Cue1, and Ubc7 with Doa10-GFP and its L8/9b truncation variant (ΔL8/9b). Doa10, Ubc6, Cue1, and Ubc7 were overexpressed from CEN/ARS plasmids using a *GAL1* promoter. **h**, **i** Bpa was introduced to indicated positions in Doa10, and UV photocrosslink to Ubc6 (panel **h**) or Cue1 (panel **i**) was examined by immunoprecipitation and immunoblotting. Ubc6, Cue1, and Ubc7 were overexpressed from CEN/ARS plasmids using a *GAL1* promoter while Doa10 was expressed from CEN/ARS plasmids using a *TDH3* promoter. Data in (**g**, **h**, **i**) are representative of two independent experiments. Source data are provided as a Source Data file.

simulations performed with Doa10 alone (Supplementary Fig. 4c). Thus, E2 and Ub seem to have some stabilizing effect on the position of RING-CH. Interestingly, RING-CH showed higher flexibility in the Doa10–Cue1–Ubc7–Ub complex than in the Doa10–Ubc6–Ub complex (Supplementary Fig. 11b; note broader distributions of the distance in

Doa10–Cue1–Ubc7–Ub). In combination with a longer distance from the C-terminus of Ub to the tunnel entrance, this aspect might contribute to the efficiency in the Ub chain elongation reaction by Ubc7[31,39]. Taken together, our computational analyses suggest that the binding of a substrate polypeptide to the lateral tunnel would

optimally position the substrate for ubiquitination while flexibility of the RING-CH and E2 domains would allow elongation of a poly-Ub chain.

## Substrate access to the middle-domain tunnel

The well-characterized Deg1 degron provided us with an opportunity to examine requirements in recognition of membrane protein substrates by Doa10. Previously, it has been shown that an insertion of two-spanning membrane protein Vma12 between Deg1 and Ura3 of the Deg1-Ura3 substrate renders much more efficient degradation of the substrate protein[37] (Fig. 8a, left panel). The Vma12 portion localizes the protein directly into the ER membrane, and this would increase the frequency of the encounter between Deg1 and Doa10. In our growth assay using Doa10 mutants, we could recapitulate this enhanced degradation of the substrate protein (and thus lesser extents of a growth rescue by mutations) upon Vma12 insertion (Supplementary Fig. 12a).

Our model proposing recognition of the Deg1 degron by the lateral tunnel in the middle domain predicts that the Deg1 portion of Deg1-Vma12-Ura3 would also need to access the tunnel interior for efficient polyubiquitination. Because the two TMs of Vma12 must remain in the bulk membrane outside Doa10 while the Deg1 degron interacts with the tunnel, we hypothesized that a certain distance (~20–30 Å) between Deg1 and the first TM of Vma12 is required for Deg1 recognition. In fact, in the original Deg1-Vma12-Ura3 fusion protein[37], there is an ~130-amino-acid-long segment that can potentially span ~100 Å between the C-terminus of Deg1 and the first TM of Vma12 (Fig. 8a, left panel). To test our hypothesis, we truncated different lengths in the C-terminal region of Deg1 and the N-terminal portion of Vma12 and performed the uracil-dependent yeast growth assay (Fig. 8a, b). Truncating the last 32 amino acids of Deg1 (Deg1$^{1-35}$-Vma12$^{FL}$) or the first 80 amino acids of Vma12 (Deg1$^{1-67}$-Vma12$^{\Delta 80}$) caused little effects on the growth phenotype. However, trimming the C-terminus of Deg1 in combination with a Vma12 truncation (Deg1$^{1-35}$-Vma12$^{\Delta 80}$) partially rescued the Doa10-dependent

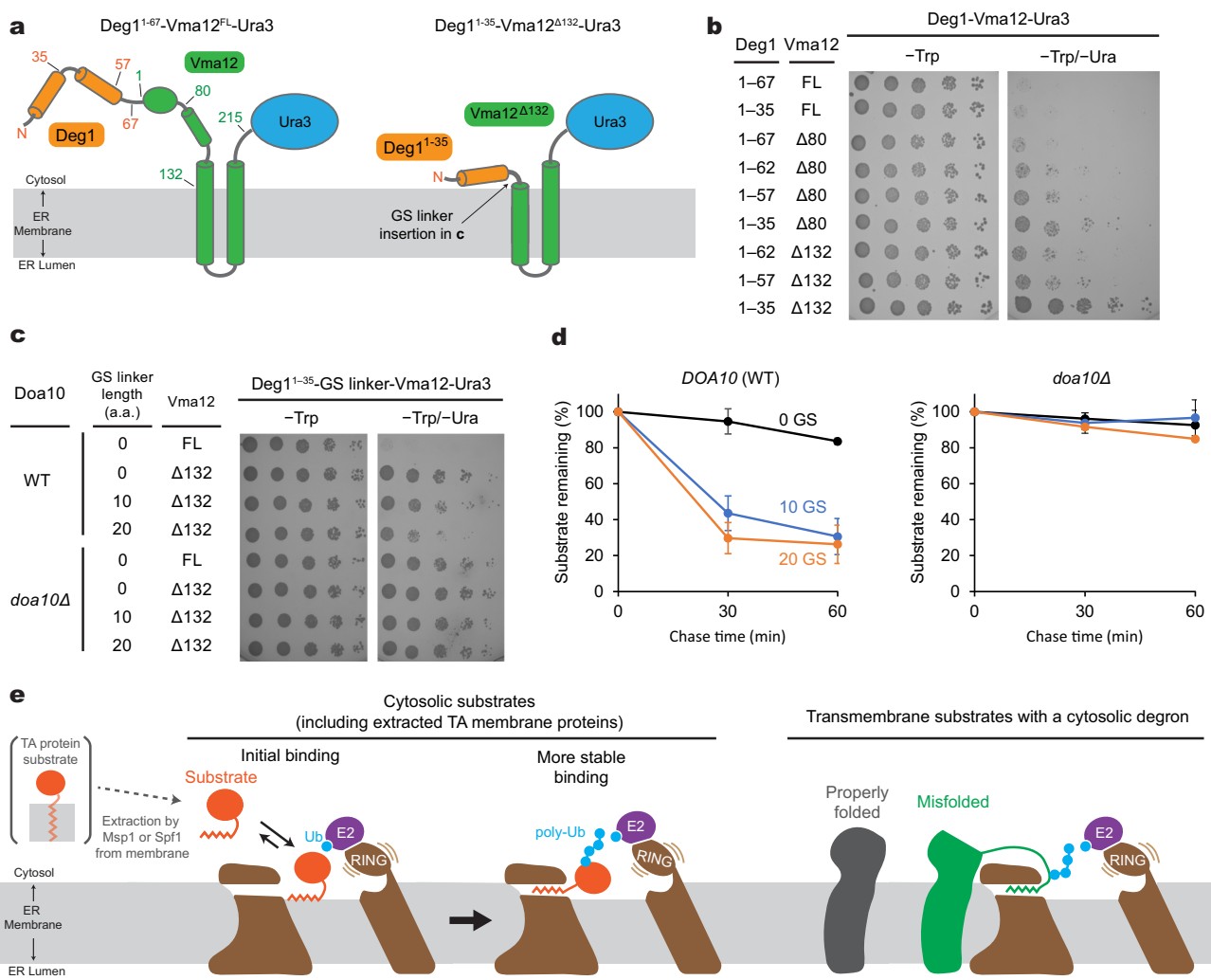

**Fig. 8 | Physical constraints in recognition of a membrane-protein-linked Deg1 degron by Doa10 and working model for substrate recognition and poly-ubiquitination. a** Schematic diagram of the Deg1-Vma12-Ura3 model substrate. Left, original construct; right, minimal construct. Numbers indicate the amino acid positions. **b** Doa10-dependent yeast growth inhibition using various truncation variants of Deg1-Vma12-Ura3. **c** As in (**b**), but a Gly/Ser (GS) linker was inserted between the truncated Deg1 (Deg1$^{1-35}$) and Vma12$^{\Delta 132}$. Two different GS linkers were tested: '10 GS' = GGS GGS GGS G and '20 GS' = GGS GGS GGS GGS GGS GGS GG. **d** Degradation of indicated Deg1$^{1-35}$-GS-Vma12$^{\Delta 132}$ variants were measured by cycloheximide chase and immunoblotting (also see Supplementary Fig. 12b). Means and s.e.m. of three independent experiments. **e** Working model for the substrate-recognition mechanism of Doa10. Cytosolic substrates could be either soluble proteins with an N- or C- terminal degron (e.g., Deg1 and CL1) or TA membrane proteins that are extracted by the Msp1 and Spf1 ATPases (e.g., Pex15Δ30). Transmembrane substrates could be membrane proteins with a degron (either intrinsic or generated by damage or unfolding) in their cytosolic domains (e.g., Deg1-Vma12). Data in (**b**, **c**) are representative of two independent experiments. Data in (**d**) are mean ± s.e.m. from three independent experiments.

growth impairment, suggesting that they became less efficient substrates to Doa10. With the complete deletion of the segment of Vma12 preceding the first TM and a minimal Deg1 sequence (Deg1[1–35]-Vma12[Δ132]), Doa10-dependent degradation of Deg1-Vma12-Ura3 could be almost completely blocked (Fig. 8b). When we reintroduced a flexible Gly/Ser-linker into the truncated region, the degradation was substantially restored both at the steady-state (Fig. 8c) and kinetic levels (Fig. 8d; Supplementary Fig. 12b). An incomplete rescue by the Gly/Ser-linker compared to full-length Vma12 (Deg1[1–35]-Vma12[FL]) might suggest some negative effects of the Gly/Ser-linker or positive effects of the N-terminal segment of Vma12 on substrate ubiquitination. Overall, these data support our model that in the case of membrane-embedded protein substrates, the degron signal needs to be extended out from the membrane domain to reach the lateral tunnel from the periphery of Doa10.

## Discussion

Doa10 possesses a remarkably broad array of substrates, including soluble proteins and integral membrane proteins, and yet, the molecular mechanism by which Doa10 recognizes its substrates remained unclear. Based on the data presented here, we propose the following model to illustrate how Doa10 binds and polyubiquitinates its substrates (Fig. 8e). Our crosslinking experiments showed that Doa10 recognizes different classes of substrates, such as Deg1, CL1, Sbh2 and Pex15Δ30, through a direct interaction within its lateral tunnel formed in the conserved middle domain. This interaction positions the substrates right above the central cavity, which is necessary for efficient polyubiquitination. Many Doa10 substrate characterized so far contain a linear amphipathic (e.g., Deg1 and CL1) or hydrophobic motif (e.g., Sbh2 and Pex15Δ30) at their N- or C- terminus, and these structural features have been shown to be critical for the Doa10 recognition[24,28,41–46]. Thus, it is likely that substrate recognition is mediated by hydrophobic interactions between the lateral tunnel and hydrophobic features of the degrons. In addition, prior to insertion into the lateral tunnel, these hydrophobic degrons presumably peripherally associate first with lipids in the central cavity. Such lipid-mediated recruitment of substrates would increase the efficiency of substrate binding to the lateral tunnel as the tunnel is rather concealed near the cytosol-membrane interface. Once the substrate is stably bound, the flexible RING-CH domain and tethered E2s would transfer Ub molecules onto the substrate polypeptide above the central cavity.

Although more substrates of Doa10 remain to be discovered, our current data suggest that the lateral tunnel can possibly be used for a broad range of substrates, including TA membrane proteins. Recently, it has been found that mistargeted TA proteins (e.g., Pex15Δ30) become substrates of Doa10 after extraction from the mitochondrial outer membrane by Msp1 or from the ER membrane by Spf1 (ref. 34,35,47). Once extracted into the cytosol by these ATPases, these TA transmembrane helices might first peripherally associate with the ER membrane and then be recruited to the central cavity and ultimately to the lateral tunnel of Doa10. It has also been shown that certain unimported mitochondrial matrix proteins are targeted for degradation by Doa10 in addition to two other E3 ligases San1 and Ubr1 in a manner dependent on the presence of an N-terminal mitochondrial-targeting sequence (MTS)[48]. Given the notion that MTSs typically form an amphipathic α-helix, akin to Deg1, it is tempting to speculate that recognition of these proteins by Doa10 might also be mediated by an interaction between their MTS and the lateral tunnel of Doa10.

While our data suggest that the lateral tunnel serves as a major recognition site for substrates of Doa10, it remains to be elucidated whether Doa10 also uses alternative mechanisms for substrate recognition. A previous study showed that Sbh2 (a paralog of the Sec61β subunit in yeast) is an ERAD-M substrate of Doa10, where its degradation requires the intact transmembrane helix and the following short ER-luminal C-terminal tail[28]. Our data showing that tunnel mutants of Doa10 are still substantially functional toward the degradation of

Sbh2 suggest that Sbh2 may potentially use an alternative recognition mechanism in addition to the lateral tunnel. In this scenario, an ER membrane-embedded form of Sbh2 might interact with Doa10 for polyubiquitination through an alternative substrate-binding site, possibly within the membrane domain of Doa10, without prior extraction.

Another example of ERAD-M-type substrates is Ubc6, which is also a TA protein. Besides functioning as an E2 enzyme for Doa10, it has been shown that Ubc6 can act as a substrate of Doa10 (ref. 27). Like Sbh2, it is unclear how Ubc6 is recognized by Doa10 as an ERAD substrate, but it has been shown that a mutation of E633, one of lateral tunnel residues, to aspartic acid greatly reduces the turnover rate of Ubc6 without significantly affecting turnover rates of Deg1 substrates[38]. A more recent study showed that Doa10 also performs a retrotranslocase activity to extract Ubc6 from the membrane independently of Cdc48 (ref. 39). These observations suggest that Doa10 might also use its central cavity and lateral tunnel but potentially in a somewhat different manner for ERAD-M substrates.

In addition to the above-mentioned examples, peripheral membrane proteins, such as the squalene monooxygenase Erg1 and lipid-droplet phospholipase Pgc1, are also on the list of Doa10 substrates[42,49]. The AlphaFold2 models of Erg1 and Pgc1 suggest that they would peripherally associate with the cytosolic leaflet of the membrane through their C-terminal hydrophobic helices. Since their structures seem to be overall well-folded without an extended tail like Deg1, their degradation is less likely to be dependent on Doa10's lateral tunnel. Instead, it is possible that Erg1 and Pgc1 may simply enter the area on Doa10's central cavity from the bulk membrane and become polyubiquitinated. We note that from our cryo-EM structure, the cytosolic leaflet is continuous between the bulk membrane and the central cavity through a space above TMs 1 and 3. Future investigations would be necessary to understand how such peripheral membrane protein substrates are recognized by Doa10 and how the central cavity and lateral tunnel are involved in the process.

Lastly, one highly intriguing feature of Doa10 is the C-shaped circular topology, where the N-terminal RING-CH domain co-folds with the CTE at the joint. The topology creates a closed, lipid-filled central cavity. AlphaFold2 models of metazoan MARCHF6 and plant SUD1 show that this feature is universal in the Doa10 homologs. Previously, it has been shown that the CTE is critical for Doa10-mediated turnover of most substrates[37,50]. Why did the RING-CH domain of Doa10 evolve to require partnering with the CTE for its function? One potential function of the CTE might be to restrict the position of the RING-CH domain toward the central cavity. Without additional anchoring through TM14, the RING-CH domain might be too flexible for efficient substrate ubiquitination. Another possibility is to tie the closed circular topology with the RING-CH domain's activity to prevent other ER integral membrane proteins from accidentally entering the central cavity and being polyubiquitinated. We attempted to create circularly permuted versions of Doa10 that maintain the polyubiquitination function to further test the latter hypothesis, but so far, this effort has been unsuccessful. While our results from the Deg1-Vma12-Ura3 experiments indirectly suggest that the closed circular topology of Doa10 can act as a barrier for integral membrane proteins, a full understanding of the physiological roles of the circular topology warrants further biochemical studies.

## Methods

### Yeast strains and plasmids

Yeast strains and plasmids used in this study are listed in Supplementary Tables 2 and 3.

To enable purification of Doa10 from yeast, we modified the yeast strain BY4741 by inserting a sequence encoding a cleavable GFP-tag between the last amino acid and stop codons of the chromosomal copy of Doa10. The final expressed protein from the resulting strain ySI-118 has an amino acid sequence of (N-terminus)…ENLPDES

(Doa10)- AGGATTASGTG (linker)- ENLYFQG (a tobacco etch virus [TEV] protease site)- TASGGGS (linker) KGEELF…(GFP)-(C-terminus). A sequence containing the TEV-GFP-tag and a nourseothricin resistance marker module was amplified from pSK-B399-GFP-NAT (a gift from the Klinge lab) with primers containing homologous arms to the chromosomal insertion site (forward primer: acg agg ttt aca cta agg gta gag ctt tag aaa att tac cag atg aaa gtG CTG GAG GGG CTA CCA CG; reverse primer: aac ata taa ctt aat gta gat ata tat atg taa ata tgc tag cat tca ttG GCC GCA TAG GCC ACT AG. Lowercase for a homologous sequence to the Doa10 sequence, and uppercase for binding to the plasmid pSK-B399-GFP-NAT). The amplicon was transformed to the BY4741 strain using standard lithium acetate protocol and plated on a YPD agar plate (1% yeast extract, 2% peptone, 2% glucose, and 2% bacto-agar) supplemented with 100 μg ml$^{-1}$ nourseothricin. Single colonies were grown with a YPD medium under nourseothricin selection, and correct insertion was verified using PCR of genomic DNA and by Sanger sequencing.

To overexpress the GFP-tagged Doa10, the endogenous promoter was replaced with a LEU2::P$_{GAL1}$ cassette that was assembled using overlap extension PCR. The LEU2 expression cassette was amplified from pYTK075 (forward primer: gct aag ata atg ggg TCG AGG AGA ACT TCT AGT ATA TCT AC, lowercase for a homologous sequence to GAL1, uppercase for binding to 5' end of the LEU2 gene. reverse primer: gat ttc aaa aac tgt ttt ttt agc caa gag tac cac taa ttg aat caa agC TGC CTA TTT AAC GCC AAC, lowercase for a homologous sequence to the chromosomal sequence flanking Doa10's promoter and uppercase for binding to LEU2 gene). P$_{GAL1}$ was amplified from pYTK030 (Forward primer: aga agt tct cct cga CCC CAT TAT CTT AGC CTA AAA AAA C, lowercase for a homologous sequence to LEU2, uppercase binds to 5' end of the GAL1 gene; reverse primer: tgc agt tca tct ctt aac ctg gag aca tta acg tca gaa tca aca tcc atT ATA GTT TTT TCT CCT TGA CGT TAA AGT ATA G, lowercase for a homologous sequence to the chromosomal sequence flanking Doa10's promoter and uppercase for binding to GAL1 gene). pYTK plasmids are part of MoClo Yeast Tool Kit[51]. The two PCR products were purified and amplified as overlapping PCR with the terminal primers. The final amplicon was transformed into yeast strain ySI-118, and selected on SC(−Leu) agar medium containing 2% glucose. Single colonies were selected, and proper integration was verified using PCR of genomic DNA. The resulting strain is designated ySI-154.

To generate Doa10-expressing plasmids for mutational and crosslinking studies, we adapted a Golden Gate cloning strategy[52]. Full-length DOA10 nucleotide sequence is known to be toxic to E. coli cells, preventing propagation of plasmids containing the coding sequence (CDS) of DOA10 (ref. 53). To overcome this issue, we splitted the CDS of DOA10 into two plasmids, which did not exhibit growth inhibition in E. coli cells. The first plasmid contains a promoter for Doa10 expression (either the endogenous DOA10 promoter, a GAL1 promoter, a RET2 promoter or a TDH3 promoter), the first 1824 bp of the CDS of DOA10, two BsaI endonuclease restriction sites, and a CEN6-ARS4 module, in this order. The second plasmid contains a BsaI site, 1822th to 3957th bp of the CDS of DOA10 and a C-terminal tag (either GFP or ALFA-tag), an ENO1 terminator sequence, and LEU2 marker cassette, and a second BsaI site. The full-length Doa10-encoding plasmid was formed by joining the two plasmids using BsaI Golden Gate cloning prior to transformation into yeast cells. Transformation of yeast with each split plasmid alone does not form colonies on a Leu drop-out agar medium because of separation of the LEU2 marker and CEN6-ARS4. Only the full plasmid joined by a BsaI Golden Gate reaction, which encodes a full length Doa10 CDS, can be stably maintained in yeast cells. Proper expression of Doa10 proteins were confirmed by immunoblotting analysis of whole cell lysates of transformants. We note that the DNA sequence for the first 322 amino acid residues of Doa10 in the first plasmid originated from gene synthesis of a reverse translated sequence and thus contains many silent mutations.

For the yeast growth (spot) assays using Deg1-Ura3 as a substrate, yeast strain MHY4086 (doa10Δ::hphMX4, lys2-801::LYS2::Deg1-Ura3) was used[54]. Plasmids with various Doa10 mutants were generated by the Golden Gate cloning approach as described above and transformed into MHY4086. Doa10 was expressed either from an endogenous promoter (P$_{DOA10}$) or RET2 promoter (P$_{RET2}$). Colonies were selected on a −Leu drop-out synthetic complete medium (SC[−Leu]) containing 2% glucose.

For the spot assays using Deg1-Vma12-Ura3 as a substrate, yeast strain MHY10818 (doa10Δ::hphMX; ref. 50). The strain was first transformed with plasmid p414-Deg1-Vma12-Ura3 (ref. 33), which constitutively expresses Deg1-Vma12-Ura3 under a MET25 promoter and uses a Trp auxotroph marker for selection. The resulting strain was transformed with a plasmid expressing a Doa10 variant as described above. Colonies were selected on SC(−Trp/−Leu) agar medium containing 2% glucose. All the truncated and GS linker versions of Deg1-Vma12-Ura3 were generated by PCR using the plasmid p414-Deg1-Vma12-Ura3 as a template. The original plasmid p414-Deg1-Vma12-Ura3 contains a FLAG-tag between Deg1 and Vma12, which was removed in the truncated versions. To detect the substrate, we attached a FLAG-tag to the C-terminus of Ura3. All these plasmids were then transformed into either JY103 (wild-type DOA10) or MHY10818 (doa10Δ).

For the cycloheximide chase experiment using Deg1-Ura3 as a substrate, the yeast strain yKW-283 (doa10Δ::natMX; leu2::P$_{PGK1}$-Deg1-Ura3-2xStrep::hphMX4) was used. This strain was made from ySI-167 (doa10Δ::natMX) by integrating P$_{PGK1}$-Deg1-Ura3-Strep into the leu2 locus. Strain ySI-167 was made by deleting chromosomal DOA10 gene (doa10Δ::natMX) in BY4741 via transformation of a PCR product, which was amplified from pSK-B399-GFP-natMX (forward primer: gat ttc aaa aac tgt ttt ttt agc caa gag tac cac taa ttg aat caa agC TGT TTA GCT TGC CTC GTC C; reverse primer: aac ata taa ctt aat gta gat ata tat atg taa ata tgc tag cat tca ttG GCC GCA TAG GCC ACT AG. Lowercase for a homologous sequence to the DOA10 gene and uppercase for binding to the natMX marker; pSK-B399 is a gift from the Klinge lab). The amplicon was transformed into BY4741, and selected on YPD agar plate supplemented with 100 μg ml$^{-1}$ nourseothricin. Chromosomal deletion was verified using PCR of genomic DNA. To generate yKW-283, we first generated an integration plasmid (pKW155), expressing Deg1-Ura3-2xStrep under the PGK1 promoter, using the MoClo Yeast Tool Kit (YTK). The sequence of Deg1 and Ura3 was amplified and cloned individually into a pYTK001 entry plasmid, resulting pKW043 (pYTK001-Deg1), pKW050 (pYTK001-Ura3). Subsequently, all the part plasmids, including pYTK-011 (P$_{PGK1}$), pKW043 (Deg1), pKW050 (Ura3 CDS), pYTK-e205 (twin Strep-tag) and pYTK-061 (ENO1 transcription terminator), were assembled into pYTK-e102 (an integration plasmid targeting to the LEU2 locus with a hygromycin marker) by BsaI Golden Gate assembly, resulting pKW155. pKW155 was linearized with NotI and transformed into ySI-167. Colonies were selected on YPD agar medium supplemented with 400 μg ml$^{-1}$ hygromycin B. Plasmids with various Doa10 mutants were generated by BsaI Golden Gate cloning approach as described above, and transformed into yKW-283. Colonies were selected on SC(−Leu) agar medium containing 2% glucose.

For the site-specific photo-crosslinking experiments, yeast strain ySI-266 (doa10Δ::hphMX4 cue1Δ::natMX) was first made by deleting chromosomal CUE1 (cue1Δ::natMX) in MHY10818 via transformation of a PCR product, which was amplified from pYTK078 (natMX marker) (forward primer: cgc cat aaa gca tta caa tct acg atc gcg caa act ttt ttc ttt tgg ccC TGT TTA GCT TGC CTC GTC C; reverse primer: tta tgc gca tta tgg gca cac ttg cgt gtt ccc ggt aag cac tta agc gtG GCC GCA TAG GCC ACT AG. Lowercase for a homologous sequence to the Cue1 gene and uppercase for binding to the natMX marker). Chromosomal deletion was confirmed using PCR of genomic DNA. Subsequently, ySI-266 was transformed with SNRtRNA-pBpaRS(TRP) (ref. 55). Colonies were selected on SC(−Trp) agar medium containing 2% glucose. The

resulting strain was then transformed with a CEN/ARS plasmid with a *URA3* selection marker (pYTK-e111) which expresses various substrates, including $P_{TDH3}$-Deg1-Ura3-2xStrep (pKW70), $P_{GAL1}$-Deg1-Ura3-2xStrep (pKW266), $P_{GAL1}$-Ura3-2xStrep-CL1 (pKW265), $P_{GAL1}$-mCherry-2xStrep-Sbh2 (pKW267) and $P_{GAL1}$-mCherry-2xStrep-Pex15Δ30 (pKW268). Colonies were then selected on SC(−Trp/−Ura) agar medium containing 2% glucose. Subsequently, plasmids expressing $P_{TDH3}$-Doa10-ALFA with an amber codon mutation at various sites were generated by site-directed mutagenesis and then *Bsa*I Golden Gate cloning. Colonies were selected on SC(−Trp/−Ura/−Leu) agar medium containing 2% glucose.

For the fluorescence intensity measurement experiments, yeast strain yAT-055 (*doa10Δ::natMX leu2*::$P_{TEF1}$-mScarlet(i3)−2xStrep-Sbh2::*hphMX4*) and yAT-057 (*doa10Δ::natMX leu2*::$P_{TEF1}$- mScarlet(i3)−2xStrep-Pex15Δ30::*hphMX4*) were used. To generate these strains, we first used MoClo Yeast Tool Kit to generate integration plasmids (pKW276 and pKW277), expressing mScarlet(i3)-Sbh2 and mScarlet(i3)-Pex15Δ30, respectively, under the *TEF1* promoter. The sequence of mScarlet(i3)−2xstrep, Sbh2 and Pex15Δ30 was amplified and cloned individually into a pYTK001 entry plasmid, resulting pKW241 (pYTK001-mScarlet(i3)−2xStrep; type 3a part), pSI015 (pYTK001-Sbh2; type 3b part) and pKW022 (pYTK001-Pex15Δ30; type 3b part). Subsequently, all the part plasmids were assembled into pYTK-e102 (an integration plasmid targeting to the *LEU2* locus with a hygromycin marker) by BsaI Golden Gate assembly. The resulting integration plasmids were linearized with *Not*I and transformed into ySI-167. Colonies were selected on YPD agar medium supplemented with 400 μg ml⁻¹ hygromycin B. Plasmids, expressing various Doa10 mutants under the *RET2* promoter, were generated as described above, and transformed into yAT-055 and yAT-057. Colonies were selected on SC(−Leu) agar medium containing 2% glucose.

For co-immunoprecipitation of E2 enzymes with Doa10, a plasmid (pYC-302) expressing Cue1, Ubc7 and Ubc6 was made. The chromosomal *CUE1*, *UBC6* and *UBC7* were first amplified by PCR and cloned into pYTK-001 entry plasmids individually. The resulting plasmids were then assembled using YTK parts to add a *GAL1* promoter, an epitope-tag, and an *ENO1* terminator by Golden Gate cloning and to form the multigene-expression plasmid pYC-302 ($P_{GAL1}$-Cue1-2xFLAG | $P_{GAL1}$-Ubc7-2xSPOT | $P_{GAL1}$−3xFLAG-6xHis-Ubc6; a CEN/ARS plasmid with a Ura3 selection marker). The amino acid sequences of used tags are: Cue1-GSGGG-DYKDDDDK-DYKDDDDK (C-terminal 2xFLAG-tag), Ubc7-GSG-PDRVRAVSHWSS-GGGSGGGST-PDRVRAVSHWSS (C-terminal 2xSPOT-tag), and MDYKDDDDK-DYKDDDDK-DYKDDDDK-GHHHHHHGS-Ubc6 (N-terminal 3xFLAG-6xHis-tag).

For photo-crosslinking experiments with Cue1 and Ubc6, plasmids pYC-300 and pYC-301 were generated to express Cue1 and Ubc7 and Ubc6 respectively. The pYTK-001 plasmids coding *CUE1*, *UBC6* and *UBC7* were assembled by Golden Gate cloning to form pYC-300 ($P_{GAL1}$-Cue1-2xFLAG | $P_{GAL1}$-Ubc7-2xStrep; a CEN/ARS plasmid with a Ura3 selection marker) and pYC-301 (e111-$P_{GAL1}$−3xFLAG-6xHis-Ubc6; a CEN/ARS plasmid with a Ura3 selection marker) as described above. Yeast strain ySI-266 (*doa10Δ::hphMX4 cue1Δ::natMX*) was used for Cue1 crosslinking, while yYC-307 (*doa10Δ::hphMX4 ubc6Δ::natMX*) was made using a similar strategy to delete chromosomal *UBC6* (*ubc6Δ::natMX*) (forward primer: gac ttt aaa tat taa cta aaa ccg cat tcg caa att gca aac aaa gta cgt aca ata gta CTG TGG ATA ACC GTA GTC G; Reverse primer: tca aaa ttt atc taa agt tta gtt cat tta atg gct tca ttt cat aaa aag gcc aac caa GGG CGT TTT TTA TTG GTC. Lowercase for a homologous sequence to the *CUE1* gene and uppercase for binding to the natMX marker sequence in the YTK plasmid pYTK078).

## Purification of Doa10 protein

Yeast strain ySI-154 was inoculated into a YP-raffinose medium (1% yeast extract, 2% peptone, and 2% raffinose) to an optical density at 600 nm (OD$_{600}$) of 0.2 and grown in shaker flasks at 30 °C until OD$_{600}$ reached 0.5. Doa10 expression was then induced by adding 2% galactose to the culture, and cells were grown until OD$_{600}$ reached ~2. Cells were then pelleted, flash frozen in liquid nitrogen, and stored in −75 °C until purification.

Cell lysis was performed by cryo-milling (SPEX SamplePrep) cycling 15 times with 1 min on time and 2 min off time. Broken cells were resuspended in a buffer containing 50 mM Tris-HCl pH 7.5, 200 mM NaCl, 1 mM EDTA, 10% glycerol, 2 mM DTT, supplemented with protease inhibitors (5 μg/ml aprotinin, 5 μg/ml leupeptin, 1 μg/ml pepstatin A, and 1.2 mM PMSF). Membranes were further solubilized by the addition of 1% lauryl maltose neopentyl glycol (LMNG; Anatrace) and 0.2% cholesteryl hemisuccinate (CHS; Anatrace), and stirring for 1.5 h at 4 °C. The cell lysate was clarified by ultracentrifugation using Beckman Type 45 Ti rotor at 186,000 *g* for 1 hr. The clarified lysate was supplemented with 25 μg/mL Benzonase nuclease and incubated with home-made agarose beads conjugated with anti-GFP nanobody at 4 °C for 2.5 h by gentle rotating. The sample was transferred to a gravity column, washed with a buffer (WB) containing 50 mM Tris-HCl pH 7.5, 200 mM NaCl, 1.0 mM EDTA, 2 mM DTT, 0.02% glycol-diosgenin (GDN; Anatrace), and 10% glycerol. Bound Doa10 was eluted by incubating the beads with ~10 μg/mL TEV protease overnight. The eluate was then injected to a Superose 6 10/300 GL Increase column (GE Lifesciences) equilibrated with 20 mM Tris-HCl pH 7.5, 100 mM NaCl, 1 mM EDTA, 2 mM DTT, and 0.02% GDN. Peak fractions were concentrated to ~5.5 mg/mL using Amicon Ultra (100-kDa cutoff; GE Lifesciences) and used immediately for cryo-EM grid preparation.

## Cryo-EM analysis

The Doa10 sample was supplemented with 3 mM FFC8 (Anatrace) before freezing cryo-EM grids. 3 μL of the sample was applied on each gold Quantifoil R 1.2/1.3 holey carbon grid that was glow discharged for 35 s using a PELCO easiGlow glow discharge cleaner. The grids were blotted for 3–4 s using Whatman No. 1 filter paper at 4 °C and 95-100% relative humidity, and plunge-frozen into liquid ethane using a Vitrobot Mark IV (FEI company). Data was collected on a Krios G2 microscope (FEI company) equipped with a Gatan Quantum Image Filter (with 20 eV slit width) and a Gatan K3 direct electron detector (Gatan). The microscope was operated at an acceleration voltage of 300 kV. The magnification was set to 64,000× under the super-resolution mode with a physical pixel size of 0.91 Å. The total exposure was set to 50 electrons/Å² divided into 50 frames, and the defocus range was set between −0.8 and −2.0 μm. All the data was acquired using SerialEM software.

For detailed illustration of the data analysis, refer to Supplementary Fig. 2c. In short, two datasets, 1760 movies (Dataset 1) and 2679 movies (Dataset 2), were pre-processed first with Warp (version 1.0.9; ref. 56) to produce an initial model that was used for cryoSPARC template picking, and again in CryoSPARC (version 2.15.0; ref. 57) to produce the final set of particles for 3D reconstruction. In Warp, the movies were corrected for motion, and contrast transfer function (CTF) estimated on micrographs divided into 7 × 5 tiles, and particles were picked by Warp's BoxNet algorithm yielding 299,741 and 241,653 autopicked particles for Dataset 1 and Dataset 2, respectively. The particles in each dataset were imported to CryoSPARC for 2D classification and ab-initio reconstruction, generating three initial models. Only one of the 3 maps presented proteinaceous features and was selected as a template for CryoSPARC template particle picking. The raw movies were re-processed using CryoSPARC (tile-based motion correction, CTF estimation, and manual movie curation) and a new particle data set containing 455,702 and 613,378 particles was obtained with the template picker. The datasets were subjected to a 2D classification, and good classes were selected after visual inspection. For Dataset 1, we used the previously obtained ab-initio models to generate three heterogeneous refinement models, which were subsequently used as reference for the heterogeneous refinement of Dataset 2. A single 3D class from each data set was

selected and their corresponding particles were subjected to a second round of heterogeneous refinement with the same 3D reference models. The particles corresponding to the best class in each data set were combined, and after a CTF refinement, particle curation yielded 324,019 particles. A non-uniform refinement job (CryoSPARC version 3.0.0) on these particles resulted in the final map of Doa10 at 3.2-Å overall resolution. The enhanced map shown in Fig. 1d–e was generated using DeepEMhancer (ref. 58). The local resolution distribution is calculated in CryoSPARC.

### Atomic model building
An initial atomic model was generated de novo using Coot (version 0.9) and cryo-EM maps that were sharpened by various B-factors. The model was then refined using the real-space refinement program of the Phenix package (version 1.16; ref) and a cryo-EM map sharpened at a sharpening B-factor of −85 Å$^{-2}$. During the course of this study, AlphaFold2 was published[40]. The AlphaFold2 model (https://alphafold.ebi.ac.uk/entry/P40318) enabled us to build additional amino acids in CTD (TMs 11-14) and L8/9b, which were difficult to confidently register de novo due to their lower local resolution in our cryo-EM map. The final model was refined again with real-space refinement of Phenix (version 1.19.2). Molprobity in the Phenix package was used for the validation. Structural models for Doa10 with E2s and Ub were predicted with AlphaFold2 (version 2.1.1) using full-length amino acid sequences of the proteins. Figures of cryo-EM maps and atomic models were generated using UCSF Chimera (ref. 59), ChimeraX (ref. 60), and PyMOL (Schrödinger).

### Sequence conservation analysis
Amino acid sequences of Doa10 from various species were obtained from UniRef90 (https://www.uniprot.org). Out of 100 sequences available, 14 sequences that were shorter than 800 amino acids were removed, and the remaining 86 sequences were subjected to multiple sequence alignment using MAFFT with default parameters. Aligned sequences were opened in UCSF Chimera to map amino acid conservation onto the Doa10 structure.

### Molecular dynamics (MD) simulations
The Doa10 MD simulations were based on the hybrid atomic model described above including residues 30 to 240, 463 to 1052, and 1095 to 1318. In addition to protein, densities were observed that can be fit with 4 phosphatidylcholine molecules, a triglyceride, an ergosterol, and a cholesterol-like detergent. These were modeled with 4 POPC molecules, a triglyceride (16:0,16:1,18:0), an ergosterol, and the detergent was replaced with ergosterol. Zinc ions were added to the RING-CH domain, with coordinating cysteine residues deprotonated and no further constraints applied[61]. Otherwise, residue protonation states were assigned with H++ (ref. 62). The N terminus, starting at residue 30, and C terminus, residue 1318, were capped with neutral acetyl and amide groups, respectively. The termini for the missing loop residues were also neutral capped. This WT model was placed in a realistic yeast ER membrane consisting of 48% POPC, 20% POPE, 10% PLPI, 8% POPS, 3% POPA, 10% ERG, and 1% DYGL (ref. 63–65) using CHARMM-GUI (version 3.7 accessed May 2023; ref. 66). Double mutants (E633V/S738V, E713V/D714V) were generated by mutating the sidechains of the respective residues. Loop substitution mutants were generated by replacing 710–718 with the following sequences: GGSGGSGGS (Δ710-718::GS) or GGSVVVGGS (Δ710-718::GS+3Val). In addition to the wild type and mutant systems, the AlphaFold2 models for Doa10 complexed with either Cue1, Ubc7, and Ub or Ubc6 and Ub were also simulated. The AlphaFold2 predictions contain regions of low confidence (pLDDT<50), and those regions were eliminated and termini that are thus generated were neutral capped. Specifically, for Doa10 the same residues as in the wildtype and mutant simulations were retained, while for Cue1 residues 1 to 37, 63 to 116, and 138 to 203 and

for Ubc6 residues 1 to 172 and 184 to 250 are present. All residues in the models for the Ubc6 and Ub proteins were retained. For both models the zinc ion and lipid molecules present in the Doa10 structure described above were added. The wild type, mutant systems, and the Doa10 complexes were similarly placed in the yeast ER membrane model, hydrated using a TIP3P (ref. 67) water box, and neutralized with 0.15 M KCl. The all-atom wild type and mutant systems were ~408,000 atoms, while the complexes were larger, at ~480,000 and ~450,000 for the Doa10–Cue1–Ubc7–Ub and Doa10–Ubc6–Ub systems respectively. The CHARMM36m protein[68] and CHARMM36 lipid (ref. 69) force fields were employed in all simulations.

Each system was equilibrated in stages using the following protocol; (1) an initial minimization was performed followed by (2) relaxation of the lipid acyl chains for 1 ns with position restraints applied to all other atoms, (3) 10 ns with the protein and bound lipids restrained to their starting positions, and (4) 100 ns with only the protein backbone restrained to allow for lipid relaxation about the protein. Finally, an additional minimization (2000 steps) prior to unrestrained NPT dynamics was performed. For equilibration, NAMD 2.14 was used, while production runs were performed in duplicate for 1 μs per replica with GPU-accelerated NAMD3 (ref. 67). All simulations were performed at a constant temperature of 310 K using Langevin dynamics (damping coefficient 1/ps), a constant pressure of 1 atm using a Langevin piston barostat, and periodic boundary conditions. Following initial equilibration, hydrogen mass repartitioning was invoked, allowing for a 4-fs time step[68]. Short range non-bonded interactions were cut off at 12 Å, with a force-based switching function starting at 10 Å. Long range non-bonded interactions were calculated using particle-mesh Ewald method with grid spacing of at least 1/Å$^3$ (ref. 69). Analysis was carried out and images were rendered with VMD (version 1.9.4a51; ref. 70).

### Yeast growth assay
Overnight cultures were diluted in a five-fold serial dilution from an OD$_{600}$ of 0.1, and 5 μl each were spotted onto an indicated agar medium. Plates were incubated at 30 °C for 2–3 days before imaging.

### Cycloheximide chase assay
For the experiment in Fig. 4f, the yeast strain yKW-283 (doa10Δ leu2::P$_{PGK1}$-Deg1-Ura3-strep) was transformed with an empty vector (pYTK-e112) or plasmids encoding various Doa10 mutants under a DOA10 promoter. Cells were grown in a synthetic medium SC(−Leu). For the experiment in Fig. 8d, the yeast strains were identical with the ones used in the yeast growth assays in Fig. 8c and grown in a synthetic medium SC(−Trp).

Overnight cultures were diluted to an OD$_{600}$ of 0.2 and grown at 30 °C until the cells reached mid-log growth phase. 5 OD$_{600}$ units of cells were collected. The pellets were resuspended in 4 ml of fresh medium. 0.25 mg/ml cycloheximide was added to the yeast suspension. Transferred 950 μl of the yeast suspension to a tube containing 20x stop mix (200 mM sodium azide and 5 mg/ml BSA). Pelleted the cells by centrifugation at 6000 g for 1 min. The pellets were resuspended with 200 μl of 0.1 M NaOH and incubated at room temperature for 5 min. Subsequently, cells were spun down by centrifugation at 12,000 g for 1 min and resuspended in 50 ml of reduced SDS sample buffer. Samples were heated at 95 °C for 5 min before analysis by SDS-PAGE and immunoblotting.

### Site-specific photo-crosslinking assay
In all the substrate crosslinking experiments, the yeast strain ySI-266 was transformed with the plasmid SNRtRNA-pBpaRS(TRP), the plasmid expressing Deg1-Ura3-Strep under the TDH3 promoter, and the plasmid expressing ALFA-tagged Doa10 under the TDH3 promoter. An amber codon was introduced at a specific site using site-directed mutagenesis.

Overnight culture was diluted to an $OD_{600}$ = 0.02 in 50 ml of fresh minimal medium SC(−Trp/−Leu/−Ura) containing 2% glucose and 2 mM Bpa (Amatek, cat# A-0067). The cells were grown at 30 °C overnight until the cells reached an $OD_{600}$ of 1.0 to 1.2. Cells were harvested, washed with deionized water and aliquoted into two tubes. One was kept on ice as a control experiment, while the other sample was transferred to a 24-well plate and UV irradiated for 1 h on ice-cold water in the cold room. Cells were pelleted and resuspended in 0.5 ml of lysis buffer (buffer LB, 50 mM Tris-HCl pH 7.5, 200 mM NaCl, 10% glycerol, 2 mM DTT, 1 mM EDTA) supplemented protease inhibitors (5 µg ml$^{-1}$ aprotinin, 5 µg ml$^{-1}$ leupeptin, 1 µg ml$^{-1}$ pepstatin A, and 1 mM PMSF). Cells were then lysed by beating with 0.5-mm glass beads. Cell lysate was supplemented with 1% LMNG and 0.2% CHS and incubated at 4 °C for 1 h to solubilize membranes. Subsequently, the lysate was clarified by centrifugation at 17,000 g for 30 min. The supernatant was incubated with Sepharose beads conjugated with anti-ALFA nanobody at 4 °C for 1 h. The beads were washed three times with 1 ml of wash buffer (buffer WB, 100 mM NaCl, 20 mM Tris-HCl pH 7.5, 1 mM EDTA, 2 mM DTT and 0.02% DDM/CHS). Samples were eluted by addition of reduced SDS sample buffer and mildly heated at 37 °C for 30 min before analysis by SDS-PAGE and immunoblotting.

For crosslinking between Doa10 and E2 enzymes, yeast strain ySI-266 (for Cue1) or yYC-307 (for Ubc6) was subsequently transformed with SNRtRNA-pBpaRS, the E2-expression plasmid (pYC-300 for Cue1, pYC-301 for Ubc6), and the plasmid encoding a Doa10 amber codon mutant under the *TDH3* promoter. The cells were grown and treated with the same procedure as described above.

For urea wash controls of photocrosslinking adducts, after washing beads once with buffer WB, two additional washes were performed with buffer WB containing 6 M urea and 0.5% Triton X-100 (buffer WD). This was followed by another wash with buffer WB without urea and elution with SDS sample buffer.

For the crosslinking experiments with substrates under the *GAL1* promoter, overnight culture was diluted to an $OD_{600}$ = 0.08 in 50 ml of fresh minimal medium SC(−Trp/−Leu/−Ura) containing 2% raffinose and 2 mM Bpa. The cells were grown at 30 °C for 24 h until the cells reached an $OD_{600}$ of 0.5. Subsequently, 2% galactose was added to induce protein expression, and the cells were further incubated for an additional 4 h. The remaining steps of the procedure were identical to those described above.

## Co-immunoprecipitation
For co-immunoprecipitation of E2 with Doa10 (Fig. 7g), yeast strain ySI-167 was first transformed with pYC-302, and subsequently with an empty vector (pYTK-e112) or a plasmid expressing Doa10 (either wild-type or Δ843-883 ['ΔL8/9b']) under a *GAL1* promoter.

Overnight culture was diluted to an $OD_{600}$ of 0.2 in 10 ml of fresh minimal medium SC(−Leu/−Ura) containing 2% raffinose, grown at 30 °C until $OD_{600}$ reached 0.5, and induced with 2% galactose for 4 h. Cells were harvested, washed with deionized water, and resuspended in 200 µL buffer LB with protease inhibitors. Cells were then lysed by bead-beating. After removing the glass beads, the cell lysate was supplemented with 1% LMNG and 0.2% CHS and incubated at 4 °C for 1 h. The lysate was then clarified by centrifugation and incubated with Sepharose resins conjugated with GFP nanobody for 3 h. The resins were washed three times with buffer WB and proteins bound to the resins were eluted by SDS sample buffer.

## Fluorescence intensity measurements
Overnight culture was diluted to an $OD_{600}$ of 1.0 in 1 ml of sterile water, and this diluted culture was subsequently dispensed into individual wells of a 96-well black plates (Corning, #3915) at a volume of 100 µL per well. Fluorescence measurements were conducted using a CLARIOstar plate reader (BMG Labtech). The excitation wavelength was set at 565 nm and the emission spectrum was recorded from 590 to 615 nm. Each sample was independently replicated at least three times.

## SDS-PAGE and antibodies
To measure the expression level of Doa10, 2.0 $OD_{600}$ units of cells were collected, resuspended in 230 µl of 0.26 M NaOH and 0.13 M β-mercaptoethanol. Cell suspensions were incubated at room temperature for 5 min, then spun down by centrifugation at 6000 g for 3 min. Pellets were resuspended with 50 µl of reduced SDS sample buffer. Cell lysates were incubated at 37 °C for 30 min and clarified by centrifugation at 21,000 g for 10 min before analysis by SDS-PAGE. SDS-PAGE was performed using Tris-glycine gels, except for Fig. 5b, c and Fig. 6e, f where Bis-Tris SDS-PAGE gels were used.

Immunoblotting experiments were performed with anti-Doa10 antiserum (a gift from M. Hochstrasser; 1:1,000 dilution), anti-GFP antibody (Thermo Fisher #MA5-15256; 1:3,000 dilution), anti-ALFA-tag (ref. [71]) and anti-SPOT-tag (BC2; ref. [72]) nanobodies fused with a rabbit Fc domain (home-made), anti-Strep-tag antibody (GenScript #A00626; 1:2,000 dilution), anti-FLAG-tag antibody (Sigma #F1804; 1,1000 dilution), anti-Pgk1 antiserum (a gift from J. Thorner; 1:1,000 dilution). Secondary antibodies used in this study were goat anti-rabbit (Thermo Fisher #31460; 1:10,000 dilution), goat anti-mouse (Thermo Fisher #31430; 1:10,000 dilution).

## Reporting summary
Further information on research design is available in the Nature Portfolio Reporting Summary linked to this article.

## Data availability
The cryo-EM map and model generated in this study have been deposited in EM Data Bank (EMDB) and Protein Data Bank (PDB) under accession codes EMD-41508 and 8TQM, respectively. Source data are provided with this paper.

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

## Acknowledgements

We are indebted to M. Hochstrasser for yeast strains, plasmids, the Doa10 antiserum, and helpful discussions. We thank J. Thorner for the Pgk1 antiserum. This work was supported by the Vallee Scholars Program (E.P.), Pew Biomedical Scholars Program (E.P.), Hellman Fellowship (E.P.), Jane Coffin Childs postdoc fellowship (K.W.). J.C.G. acknowledges support from NIH (R01 GM123169). Computational resources were provided through XSEDE (TG-MCB130173), which is supported by NSF (ACI-1548562). This work also used the Hive cluster, which is supported by the NSF (1828187) and is managed by PACE at Georgia Tech.

## Author contributions

K.W., performed biochemical experiments, except for experiments in Fig. 7 and Supplementary Fig. 10e, which were performed by Y.C. S.I. purified samples for cryo-EM analysis and performed initial biochemical studies. E.P. and S.I. collected and analyzed cryo-EM data, and built atomic models. E.P. performed AlphaFold2 modeling. D.L. and J.C.G. performed MD simulations and analyses. A.T. assisted K.W. on biochemical experiments. E.P. supervised the project and wrote the manuscript with input from all authors. All authors contributed to data interpretations and manuscript editing.

## Competing interests

The authors declare no competing interests.
