## [Peer Review File · Nature Communications]

REVIEWER COMMENTS

Reviewer #1 (Remarks to the Author):

This manuscript reports the cryo-EM structure of *Saccharomyces cerevisiae* Doa10 in a detergent micelle. An interesting architecture is discovered with two smaller lipid-binding cavities proximal to a central cavity for substrate binding. Functional assays are used to test structure-based hypotheses and regions not observed in the cryo-EM density maps are modeled by using AlphaFold2, with the E2 interactions further modeled by AlphaFold2-Multimer, and validation through UV photocrosslinking and/or co-immunoprecipitation. Overall, the manuscript is a tour de force and impactful. Some concerns are listed below, including concerns of over-interpretation. Overall, the manuscript is not written as well as it should be in terms of properly referencing figures and/or showing effects described. However, the authors can probably address these concerns with textual changes.

Introduction: 'ref.' is not needed where shown.

Line 35-36: awkward grammar, probably should be 'ligases is to'

Line 129 and Fig 1b: The image doesn't look like a 'horseshoe' as described.

Line 138-139: The authors write 'The segment between TMs 3 and 4 includes two amphipathic α -helices lying flat on the luminal leaflet of the ER membrane.' This structure is in a detergent micelle, making this conclusion impossible from the data presented. The next sentence similarly over-interprets the data. If the conclusions are based on additional data in the literature, then the appropriate references are needed.

Line 142-145: The helices surrounding the cavities shown in Fig. 2a or 2b should be labeled TM1....

Supplementary Figure 4b: the authors should clarify which Doa10 region is included for the MD simulations.

Line 181 and 183: As also noted above, the structure is not solved in the ER membrane, nor with cytosol present.

Line 188: E633 should be shown on an image and that image referenced.

Line 215: The double mutant E633/S738 does not show a stronger effect in Fig. 4e contrary to what the authors wrote.

Line 230: L6/7 and TMs 5-7 need to be labeled on figs 4g-i. The effect described is not clear in the images. In particular, the reduced solvent-accessible space is not apparent from this view.

Line 238: the indicated amino acids are not shown/labeled in the figures. Also it would seem that Figs 4j-l should be referenced in this paragraph.

Line 258-259: This conclusion is an over-interpretation. More data would be needed to conclude that the trapped interactions don't occur through random collision.

Line 284: A sequence alignment should be shown for the L8/9b region, as this information is not clear in the referred to Supplementary Fig. 5 – which is also poorly labeled.

Line 301: Figure 6a should be referenced and the proposed distances indicated with the GG tail included. Moreover, information on alternative predictions should be included – how many models were generated in AlphaFold2-Multimer? Do molecular dynamics simulations indicate fixed distances or do these values vary over time?

Line 326-329: A control is missing for this interpretation, namely 1-35 Deg1 with FL Vma12.

Line 326: The effect of Deg1 1-62 with Vma delta 132 is similar to Deg1 1-62 with Vma delta 80 and these effects are observable but don't indicate a requirement. The authors seem to be overinterpreting these results, at least as written.

Line 330: The corresponding figures are not indicated, but if Figure 7c then inclusion of FL for comparison would be useful and at least for this data, a partial rescue is indicated but not complete restoration. If Supplementary Figure 9b, why is the 0 time point so low in lanes 4 and 7?

Reviewer #2 (Remarks to the Author):

This manuscript focuses on the structure of Doa10 which has a unique and unusual structure in the membrane. The work combines experimental and computational techniques, and my focus of the review will be on the computational aspect.

The computational approach and timescale are valid based on the stabilized RMSD data provided. The MD simulations add further proof of the flexibility of the RING-CH domain when Doa10 is placed in a

model ER membrane. My only concern is with the large overall RMSD shown in supplementary Figure 4b. What is causing this large RMSD from the cryoEM structure and are there any adjustments in the weird architecture of helices in the TM region?

Reviewer #3 (Remarks to the Author):

The manuscript by Wu and colleagues investigates the mechanism of ER-associated degradation by the ubiquitin ligase Doa10. Using cryo-EM and aided by a high confidence AlphaFold model the authors shows that Doa10 adopts a horseshoe like arrangement within the lipid bilayer, with the N- and C-termini coming together to assemble an antiparallel β -sheet and delimiting a large central lipid filled cavity. Their structure also showed an uncommon arrangement of loosely packed transmembrane segments defining additional cavities that appear to be filled by specific lipid molecules. In addition, the authors identified a water filled tunnel lined by highly conserved residues that was “capped” by a loop between TMDs 6 and 7 (L6/7 loop). Using mutagenesis and MD simulations they showed the importance of both the tunnel and the L6/7 loop for the function of Doa10. Photocrosslinking experiments show that this region of Doa10 interact specifically with Deg1, a well-established degron (or substrate) for Doa10. Aided by AF models and photocrosslinking experiments they convincingly show that the Deg1 binding site is well positioned in relation to the ubiquitin conjugating enzymes Ubc6 and Cue1/Ubc7 to facilitate Deg1 ubiquitination. Finally using a variety of Deg1 fusions they explore how membrane bound proteins access the substrate recognition site for ubiquitination by Doa10. This led that to propose that the horseshoe-like organization of Doa10 is important for it to achieve substrate specificity.

This an elegantly conducted and well organized study that significantly advances our understanding of Doa10. Considering the high conservation and degree of structural similarity between Doa10 its homologues, the findings presented in the manuscript are likely relevant to understand the mammalian counterpart TEB4/MARCHF6. Overall, I find the study of interest to a general readership of nature communications in particular if the issues listed below can be addressed.

My main concern is related with the general conclusion about the mechanism of substrate recognition by Doa10 while a single substrate was tested. While the model is appealing, it should be tested directly using other substrates. In particular, certain tail-anchored (TA) membrane proteins are known to be degraded by Doa10. There is experimental evidence (both in yeast and mammalian systems) that the degradation occurs post TA insertion in the ER membrane. It would be important to test if these substrates (such as Sbh2) use the same recognition mechanism. If this is not an option at the moment, claims about the general mechanism of substrate recognition should be softened and the possibility of Doa10 using multiple mechanisms for the recognition of substrates discussed.

Minor points:

- The authors should adopt the new nomenclature for MARCHF6 (instead of MARCH6)
- Lines 35-36: sentence is missing a verb (“The primary function of these E3 ligases recognize and polyubiquitinate....”)

- is the hybrid model combining experimental and AF predictions shown in any of the figures? Which ones? How does it compare to the AF2 model?

- Doa10 has a predicted mw of 151kDa. However, it appears to migrate by SDS page with an apparent mw of >180kDa. is this due to posttranslational modifications? can the authors comment?

- Lines 214-217: mutations that appear to increase the hydrophobicity of the tunnel appear to have a deleterious effect on Deg1-URA3 degradation. While for most of the mutants this is a reasonable conclusion, for S738V and Y742L this appear to be an overstatement given that these proteins are almost undetectable by western blot (Fig S6b).

- Photocrosslinking data looks very convincing but I wonder if the authors could comment on the fact that a doublet is observed with the anti-alfa tag antibody while the anti-strep tag blot shows a single band.

- Figure 6 legend should provide additional details. How are the various constructs expressed? Endogenous promoter or overexpression? What is SPOT?

Reviewer #1 (Remarks to the Author):

This manuscript reports the cryo-EM structure of *Saccharomyces cerevisiae* Doa10 in a detergent micelle. An interesting architecture is discovered with two smaller lipid-binding cavities proximal to a central cavity for substrate binding. Functional assays are used to test structure-based hypotheses and regions not observed in the cryo-EM density maps are modeled by using AlphaFold2, with the E2 interactions further modeled by AlphaFold2-Multimer, and validation through UV photocrosslinking and/or co-immunoprecipitation. Overall, the manuscript is a tour de force and impactful. Some concerns are listed below, including concerns of over-interpretation. Overall, the manuscript is not written as well as it should be in terms of properly referencing figures and/or showing effects described. However, the authors can probably address these concerns with textual changes.

We appreciate the reviewer's positive evaluation and helpful suggestions, particularly regarding figure references and potential overinterpretations. Below is our response to address the reviewer's concerns. All major changes are also highlighted in yellow in our revised manuscript.

Introduction: 'ref.' is not needed where shown.

Format "(ref. ##)" was used when the sentence ends with a number (e.g., Ubc7) to prevent having a superscripted number on another number according to the publisher's formatting rules.

Line 35-36: awkward grammar, probably should be 'ligases is to'

The review is correct. "is to" was missing. This error is now corrected.

Line 129 and Fig 1b: The image doesn't look like a 'horseshoe' as described.

Instead of 'horseshoe', we now use 'C-shaped' to describe the overall structure of Doa10.

Line 138-139: The authors write 'The segment between TMs 3 and 4 includes two amphipathic α -helices lying flat on the luminal leaflet of the ER membrane.' This structure is in a detergent micelle, making this conclusion impossible from the data presented. The next sentence similarly over-interprets the data. If the conclusions are based on additional data in the literature, then the appropriate references are needed.

In our opinion, it is generally acceptable in the field to describe the membrane boundaries based on the positions of the detergent micelle boundaries and distributions of hydrophobic amino acids. However, we also agree with the reviewer that we cannot be certain about this conclusion given that the structure was determined in detergent, not in the native membrane. We therefore softened this statement in our revised manuscript. We also now refer to a figure showing an MD simulation result of Doa10 in a model lipid membrane, which supports this conclusion.

Line 142-145: The helices surrounding the cavities shown in Fig. 2a or 2b should be labeled TM1....

We now labelled TMs 1-10 in Fig. 2 a and b.

Supplementary Figure 4b: the authors should clarify which Doa10 region is included for the MD

simulations.

This is now indicated in the figure legend.

Line 181 and 183: As also noted above, the structure is not solved in the ER membrane, nor with cytosol present.

We toned down the sentence ('seems') and described our MD simulation result with new Supplementary Fig. 6, which is consistent with these cryo-EM observations.

Line 188: E633 should be shown on an image and that image referenced.

We have labeled the position of E633 in Fig. 2b and referenced the figure in the text.

Line 215: The double mutant E633/S738 does not show a stronger effect in Fig. 4e contrary to what the authors wrote.

The double mutant E633V/S738V consistently produced a somewhat stronger effect than E633V alone in this growth assay (shown in Fig. 4e), but we agree that the difference is moderate. The reviewer might also have commented based on the left panel of Fig. 4f, where the effect by E633V/S738V is only slightly stronger compared to E633V. This difference could be due to some differences in steady-state (growth assay) vs the kinetic (cycloheximide chase) measurements. We modified our statements to reflect these aspects.

Line 230: L6/7 and TMs 5-7 need to be labeled on figs 4g-i. The effect described is not clear in the images. In particular, the reduced solvent-accessible space is not apparent from this view.

We added suggested labels to the figure. We reworked the figures to show the collapse of the tunnel better.

Line 238: the indicated amino acids are not shown/labeled in the figures. Also it would seem that Figs 4j-l should be referenced in this paragraph.

The figures now show the F637, M686, F689, and Y742 residues (also see figure legend) and Fig. 4j-l is referenced in the text.

Line 258-259: This conclusion is an over-interpretation. More data would be needed to conclude that the trapped interactions don't occur through random collision.

We now tested four additional Bpa sites (231, 698, 944, and 948; a total of six including two previously tested positions) in the middle domain that are cytosolically exposed and observed no clear crosslinks with any of these position in contrast to those in the tunnel (new Supplementary Fig. 9). These data further support our conclusion that the observed crosslinks are unlikely to be caused by random collisions.

Line 284: A sequence alignment should be shown for the L8/9b region, as this information is not clear in the referred to Supplementary Fig. 5 – which is also poorly labeled.

Although it was not feasible to generate a sequence alignment between Doa10 and MARCHF6 for this region due to low sequence identity (in fact, L8/9b is completely missing in MARCHF6), we added labels to the figure (Supplement Fig. 5) to guide readers in the structural comparison.

Line 301: Figure 6a should be referenced and the proposed distances indicated with the GG tail included. Moreover, information on alternative predictions should be included – how many models were generated in AlphaFold2-Multimer? Do molecular dynamics simulations indicate fixed distances or do these values vary over time?

We now referenced the figure panels (now Fig. 7) and indicated the position of the GG tail. We generated five models from AlphaFold2 multimer, and they all showed similar predictions. This information is now described in the figure legend.

In order to address whether the distance from the C-terminal ubiquitin double glycine tail residues to the tunnel entrance varies, we have performed additional molecular dynamics simulations for both complexes. Two replicas were simulated for 1 μ s for each complex. We added a new figure in the Supplementary Information (Supplementary Fig. 11; for distance distributions, see panel b) along with discussion in the main text. The distance is ~ 20 Å (± 5 Å) for the Doa10-Ubc6-Ub complex and ~ 25 Å (± 10 Å) for the Doa10-Cue1-Ubc7-Ub.

Line 326-329: A control is missing for this interpretation, namely 1-35 Deg1 with FL Vma12.

The requested control, Deg1(1-35)-Vma12(FL), is now included in the figure in our revised manuscript (now Fig. 8b,c).

Line 326: The effect of Deg1 1-62 with Vma Δ 132 is similar to Deg1 1-62 with Vma Δ 80 and these effects are observable but don't indicate a requirement. The authors seem to be overinterpreting these results, at least as written.

Deg1(1-62) with Vma12(Δ 132) or Vma12(Δ 80) have similar phenotypes in this growth assay, indicating that the further trimming from position 80 to 132 have no significant effect. This could be due to some secondary or tertiary structure formed within this region such that the change in actual distance between Deg1 and Vma12's first TM would not be large relative to a length provided by a segment of Deg1 36-62. However, when we further trimmed Deg1 from position 62 down to 35, we saw a substantial difference between Vma12(Δ 132) or Vma12(Δ 80). Overall, the data is consistent with the trend that the shorter the segment between Deg1(1-35) and Vma12(Δ 132) is the less efficient substrate degradation becomes. The control experiment of Deg1(1-35)-Vma12(FL) showed no significant difference from the Deg1(1-67)-Vma12(FL), indicating that Deg1(1-35) is a highly functional degron in the context of these experiments.

Line 330: The corresponding figures are not indicated, but if Figure 7c then inclusion of FL for comparison would be useful and at least for this data, a partial rescue is indicated but not complete restoration. If Supplementary Figure 9b, why is the 0 time point so low in lanes 4 and 7?

The corresponding figures are now properly indicated.

As the reviewer pointed out, the GS linker showed a partial rescue in our growth assay compared to full-length Vma12. We think this could be due to some negative effects of the Gly/Ser-linker (e.g., excessive flexibility) and/or positive effects of the N-terminal domain of Vma12 (additional Lys/Thr/Ser residues) on substrate ubiquitination. We made our description more accurate and added an additional sentence to comment on this observation.

Regarding the reviewer's last question, the weak band intensities of these substrates at time 0 indicate low steady-state cellular abundance due to rapid degradation in the presence of functional Doa10. This is also the case for other substrates (see new Fig. 6 e and f).

Reviewer #2 (Remarks to the Author):

This manuscript focuses on the structure of Doa10 which has a unique and unusual structure in the membrane. The work combines experimental and computational techniques, and my focus of the review will be on the computational aspect.

The computational approach and timescale are valid based on the stabilized RMSD data provided. The MD simulations add further proof of the flexibility of the RING-CH domain when Doa10 is placed in a model ER membrane. My only concern is with the large overall RMSD shown in supplementary Figure 4b. What is causing this large RMSD from the cryoEM structure and are there any adjustments in the weird architecture of helices in the TM region?

We thank the reviewer's positive comments.

To clarify, both the top and bottom panels use all C α atoms in the alignment. The top panel reports the RMSD after alignment to the initial configuration, while the bottom panel illustrates the contribution of the RING-CH domain and the membrane domain, using only the RING-CH domain residues or membrane domain residues in the RMSD measurement. We thank the reviewer for pointing this out and have included the explicit residues employed in the figure caption.

The large values of the RMSD of the RING-CH domain relative to the initial structure is caused by the transition of the upright conformation seen in the initial structure (Supplementary Figure 4a at 0 ns) to the conformations after relaxation of the RING domain over the central cavity (Supplementary Figure 4a at 100 ns-1 μ s). The membrane region is very stable, as illustrated by the lower, stable RMSD in Supplementary Figure 4c (blue and green curves).

Reviewer #3 (Remarks to the Author):

The manuscript by Wu and colleagues investigates the mechanism of ER-associated degradation by the ubiquitin ligase Doa10. Using cryo-EM and aided by a high confidence AlphaFold model the authors shows that Doa10 adopts a horseshoe like arrangement within the lipid bilayer, with the N- and C- termini coming together to assemble an antiparallel β -sheet and delimiting a large central lipid filled cavity. Their structure also showed an uncommon arrangement of loosely packed transmembrane segments defining additional cavities that appear to be filled by specific lipid molecules. In addition, the authors identified a water filled tunnel lined by highly conserved residues that was "capped" by a loop between TMDs 6 and 7 (L6/7 loop). Using mutagenesis and MD simulations they showed the importance of both the tunnel and the L6/7 loop for the function of Doa10. Photocrosslinking experiments show that this region of Doa10 interact specifically with Deg1, a well-established degron (or substrate) for Doa10. Aided by AF models and photocrosslinking experiments they convincingly show that the

Deg1 binding site is well positioned in relation to the ubiquitin conjugating enzymes Ubc6 and Cue1/Ubc7 to facilitate Deg1 ubiquitination. Finally using a variety of Deg1 fusions they explore how membrane bound proteins access the substrate recognition site for ubiquitination by Doa10. This led that to propose that the horseshoe-like organization of Doa10 is important for it to achieve substrate specificity.

This an elegantly conducted and well organized study that significantly advances our understanding of Doa10. Considering the high conservation and degree of structural similarity between Doa10 its homologues, the findings presented in the manuscript are likely relevant to understand the mammalian counterpart TEB4/MARCHF6. Overall, I find the study of interest to a general readership of nature communications in particular if the issues listed below can be addressed.

We appreciate the reviewer's positive evaluation and helpful suggestions. Below is our response to address the reviewer's concerns. All major changes are also highlighted in yellow in our revised manuscript.

My main concern is related with the general conclusion about the mechanism of substrate recognition by Doa10 while a single substrate was tested. While the model is appealing, it should be tested directly using other substrates. In particular, certain tail-anchored (TA) membrane proteins are known to be degraded by Doa10. There is experimental evidence (both in yeast and mammalian systems) that the degradation occurs post TA insertion in the ER membrane. It would be important to test if these substrates (such as Sbh2) use the same recognition mechanism. If this is not an option at the moment, claims about the general mechanism of substrate recognition should be softened and the possibility of Doa10 using multiple mechanisms for the recognition of substrates discussed.

To address this concern, we extended our study to a few other known substrates of Doa10, including CL1, Sbh2 and Pex15 Δ 30 (new Fig. 6). CL1, similar to Deg1, represents a model ERAD-C degron (but located at the C-terminus of the fusion protein), whereas Sbh2 and Pex15 Δ 30 are the TA membrane proteins. We performed Bpa photo-crosslinking experiment and tested effects of tunnel mutants on substrate degradation. All three substrates are crosslinked to Bpa incorporated to the tunnel interior (R734), suggesting that these substrates also interact with the lateral tunnel of Doa10. Consistent with this conclusion, the tunnel-collapse mutants (E633V/S738V and E713V/D714V) also show defects in the degradation of the substrates albeit to different extents. We have now included this new data in our revised manuscript (new Fig. 6) and generalize our model. Because we cannot exclude the possibility of Doa10 using alternative mechanisms for substrate recognition, particularly for Sbh2, where we observed a rather moderate defective phenotype with the strong tunnel mutant E713V/D714V, we also discussed such a possibility.

Minor points:

- The authors should adopt the new nomenclature for MARCHF6 (instead of MARCH6)

We have revised this according to the reviewer's suggestion.

- Lines 35-36: sentence is missing a verb ("The primary function of these E3 ligases recognize and polyubiquitinate....")

“is to” was missing by mistake. This error is now corrected.

- is the hybrid model combining experimental and AF predictions shown in any of the figures? Which ones? How does it compare to the AF2 model?

The hybrid model was used only for MD simulations. Except for AlphaFold2 prediction and MD simulation figures, all figures were generated with a model that was built based on the cryo-EM structure. The hybrid model simply includes additional regions, such as the RING-CH domain, that we built additionally based on the AlphaFold2 model (AF-P40318-F1). We included this hybrid model in our Supplementary Information (Supplementary Data 1) as it could be helpful to some readers.

- Doa10 has a predicted mw of 151kDa. However, it appears to migrate by SDS page with an apparent mw of >180kDa. is this due to posttranslational modifications? can the authors comment?

We do not have a good answer to this question, but to our knowledge, there is no known large PTM (such as glycosylation) in Doa10. But transmembrane proteins often migrate in SDS-PAGE quite differently from their nominal molecular weights, and thus we believe that this is due to a peculiarity of this membrane protein in interacting with SDS molecules and packing in SDS micelles (perhaps unusually long TM helices of Doa10 could be responsible).

- Lines 214-217: mutations that appear to increase the hydrophobicity of the tunnel appear to have a deleterious effect on Deg1-URA3 degradation. While for most of the mutants this is a reasonable conclusion, for S738V and Y742L this appear to be an overstatement given that these proteins are almost undetectable by western blot (Fig S6b).

Regrettably, the previous image seemed unevenly developed in the immunoblotting experiment (perhaps due to suboptimal loading and transfer). We repeated this experiment, and all the variants showed comparable expression levels.

- Photocrosslinking data looks very convincing but I wonder if the authors could comment on the fact that a doublet is observed with the anti-alfa tag antibody while the anti-strep tag blot shows a single band.

We do not think the second (upper) band is a crosslink specific to the expressed substrate. Thus, they are not technically doublet. Those bands in the ALFA blot indicated with “<” in Fig. 5 f and g could identify as the “Doa10 x Subst.” species because they nicely overlay with the band in the Strep blot. It is possible that those UV-dependent but substrate-independent bands might have originated from crosslinking to endogenous substrates or self-crosslinking to another position in the same Doa10 molecule (due to formation of a lasso-like topology, such species could have slower migration in SDS-PAGE than the uncrosslinked species).

- Figure 6 legend should provide additional details. How are the various constructs expressed? Endogenous promoter or overexpression? What is SPOT?

The information was previously provided in the Methods and Supplementary Information, but we now also briefly included in the figure legend.

SPOT-tag is a peptide affinity tag (PDRVRAVSHWSS) developed by ChromoTek that binds to the BC2 nanobody. The sequence and antibody information is now included in the Methods.

REVIEWERS' COMMENTS

Reviewer #1 (Remarks to the Author):

My concerns have been fully addressed by the authors.

Reviewer #2 (Remarks to the Author):

Changes made make this acceptable for publication.

Reviewer #3 (Remarks to the Author):

The new data (on Sbh2 and Pex15D30) and the textual changes improved greatly the revised manuscript. My earlier concerns have been addressed in full and I strongly support the manuscript in its current form. This study will be an important addition to the community interested in protein homeostasis.